



# Coupling the ParFlow Integrated Hydrology Model within the NASA Land Information System: A case study over the Upper Colorado River Basin

[1,4,7]Peyman Abbaszadeh, [3]Fadji Zaouna Maina, [1,4]Chen Yang, [5]Dan Rosen, [3]Sujay Kumar, [6]Matthew Rodell, [1,2,4]Reed Maxwell

[1]*Department of Civil and Environmental Engineering, Princeton University, Princeton, NJ, USA*
[2]*High Meadows Environmental Institute, Princeton University, Princeton, NJ, USA*
[3]*Hydrological Sciences Laboratory, NASA Goddard Space Flight Center, Greenbelt, MD, USA*
[4]*Integrated GroundWater Modeling Center, Princeton University, Princeton, NJ USA*
[5]*Climate & Global Dynamics Lab, The National Center for Atmospheric Research, Boulder, Colorado, USA*
[6] *Earth Sciences Division, NASA Goddard Space Flight Center, Greenbelt, MD, USA*
*now at [7]Department of Civil and Environmental Engineering, Hydrologic Modeling and Assimilation Lab, Portland State University, Portland, OR*
*Corresponding author: Peyman Abbaszadeh, pabbaszadeh@princeton.edu, pabbaszadeh@pdx.edu*

## Abstract

Understanding, observing, and simulating Earth's water cycle is imperative for effective water resource management in the face of a changing climate. NASA's Land Information System (LIS)/Noah-MP and the ParFlow groundwater model are the two widely used modeling platforms that enable studying the Earth's land surface and subsurface hydrologic processes, respectively. The integration of ParFlow and LIS/Noah-MP models and harnessing their strengths can provide an opportunity to simulate surface terrestrial water processes and groundwater dynamics together while enhancing the accuracy and scalability of hydrological modeling. This study introduces ParFlow-LIS/Noah-MP (PF-LIS/Noah-MP), which is an integrated, physically based hydrologic modeling framework. PF-LIS/Noah-MP enables the user to simulate land surface processes in conjunction with subsurface hydrologic processes while considering the interactions between the two. In this study, we compared the results of the coupled PF-LIS/Noah-MP and standalone LIS/Noah-MP models with a suite of in-situ and satellite observations over the Upper Colorado River Basin (UCRB) in the United States. This analysis confirmed that integrating ParFlow with LIS/Noah-MP not only enhances the capability of LIS/Noah-MP in estimating land surface processes over regions with complex topography but also enables it to accurately simulate subsurface hydrologic processes.

Keywords: ParFlow, LIS/Noah-MP, PF-LIS/Noah-MP, hydrology model, groundwater





## 1. Introduction


The interaction of surface and subsurface hydrologic processes is complex and dynamic.
Surface hydrologic processes include the movements of water on the land surface, such as runoff,
while subsurface hydrologic processes include the movements of water below the ground, such as
infiltration and groundwater flow. These surface and subsurface physical processes are
interconnected through various mechanisms. For instance, precipitation that falls on the land
surface can infiltrate the soil and become soil moisture or runoff into nearby streams and rivers.
Soil moisture can either return to the atmosphere through evapotranspiration or percolate into the
subsurface, replenishing groundwater storage. Streams and rivers can also recharge underlying
groundwater aquifers, and groundwater can discharge into rivers and streams (Fleckenstein et al.,
2010; Kalbus et al., 2006; Kourakos et al., 2019; Ntona et al., 2022; Winter et al., 1998).
The interaction of surface and subsurface hydrologic processes is particularly relevant to
managing water resources in arid and semi-arid regions, where water resources are often limited
(Deb et al., 2019; Scanlon et al., 2012; Tian et al., 2015; Wada et al., 2010). Climate change can
impact surface and subsurface hydrologic processes and their interactions and feedback to the
atmosphere. In particular, changes in precipitation patterns, temperature, and evapotranspiration
rates can affect the balance and feedback between surface water and groundwater, affecting water
availability and quality (Alley, 2007; Christensen et al., 2004; Oki and Kanae, 2006; Scanlon et
al., 2012). Besides, human activities, such as irrigation and water pumping, can alter the natural
behavior of surface–subsurface interactions (Boucher et al., 2004; Gordon et al., 2005; Leng et al.,
2014; Leung et al., 2011; Liang et al., 2003; Sacks et al., 2009; Tang et al., 2007; Tian et al., 2015),
affect the land-atmosphere coupling (Harding and Snyder, 2012; Kawase et al., 2008; Lo and
Famiglietti, 2013; Qian et al., 2013) and compromise the health of ecosystems and water quality
(Green et al., 2011; Jasechko et al., 2017; Scanlon et al., 2012).
Irrigation water use in the Upper Colorado River Basin (UCRB) is a substantial and
growing demand on the region's limited water resources. UCRB includes parts of Colorado,
Wyoming, Utah, and New Mexico and is home to a large agricultural sector. The region's irrigated
agriculture mostly relies on groundwater (Hutson et al., 2004; Kenny et al., 2005). Studies show
that due to the recent prolonged drought across the western US (Cook et al., 2015, 2021; Williams
et al., 2022), water managers have increased their dependence on groundwater to secure public
water supply and irrigate agricultural lands (Famiglietti et al., 2011, 2013; Taylor et al., 2013).



Groundwater pumping is an important source of water for agriculture in the UCRB, particularly
when and where surface water availability is limited (Castle et al., 2014). Excessive pumping can
lead to the depletion of aquifers, impacting water availability and the long-term sustainability of
agricultural practices. To address these challenges, many states in the UCRB have implemented
regulations and policies to manage groundwater use in agriculture, such as implementing
groundwater monitoring programs and setting limits on the amount of water that can be pumped
(Supplemental Environmental Impact Statement for Near-term Colorado River Operations; U.S.
Department of the Interior, 2021). In general, water management strategies can benefit from
skillful hydrologic modeling that considers the land surface and subsurface physical processes in
a coupled fashion. In this work, we introduce and test a coupled land surface-subsurface hydrology
model (hereafter integrated hydrologic model) as one means to address this need.

79        Integrated hydrologic models have been highly successful in a broad range of watershed-

scale studies (see Table 1 in Maxwell et al., 2014). These models represent observed surface and
subsurface behavior, diagnose stream–aquifer and land–energy interactions, and enhance our
understanding of how disturbances like changes in land-cover and human-induced climate change
affect different layers of the hydrologic system (Maxwell et al., 2015). The importance of the
interactions between groundwater and surface water and the use of integrated hydrologic models
to better understand this connection has been the subject of many studies in the past decade
(Barthel and Banzhaf, 2016; Brookfield et al., 2023; Kuffour et al., 2020; Lahmers et al., 2022;
O'neill et al., 2021a; Wang and Chen, 2021; Yang et al., 2021). Until recently, integrated
hydrologic models were mainly used at local to regional scales, as their implementation required
extensive computational resources. However, recent advances in parallel High-Performance
Computing (HPC) techniques, numerical solvers, and observational data have made it feasible to
conduct large scale, high-resolution simulations of the terrestrial hydrologic cycle (Kollet et al.,
2010; Maxwell, 2013; Maxwell et al., 2015; Naz et al., 2023). This has opened up new possibilities
for the practical application of integrated hydrologic models at regional to continental scales. Most
previous large-scale subsurface studies have not accounted for surface processes explicitly (Fan et
al., 2007, 2013; Miguez-Macho et al., 2007). Similarly, many continental to global-scale surface
hydrology studies have ignored groundwater or used a highly simplified model, despite the
importance of lateral groundwater flows (Krakauer et al., 2014). This limitation has been observed
in studies such as those conducted by Döll et al. (2012), Maurer et al. (n.d.), and Xia et al. (2012).




The NASA Land Information System (LIS) is a software framework designed to facilitate
the integration of land surface models and satellite remote sensing data for improved understanding
and prediction of land surface processes (Kumar et al., 2006, 2008a; Peters-Lidard et al., 2007).
LIS has been widely used for a variety of scientific and practical applications, including drought
monitoring and prediction, water resource management, and flood forecasting, among others
(Crow et al., 2012; Getirana et al., 2020; Li et al., 2019; Mocko et al., 2021; Nie et al., 2022). LIS
has been integrated with other Earth system modeling systems. For example, a coupled high
resolution land-atmosphere system has been developed by coupling LIS with the Weather
Research and Forecasting (WRF) model (Kumar et al., 2008a). This coupled land-atmosphere
system facilitates study of the interactions between the atmosphere and land surface processes.
ParFlow is a robust and versatile groundwater model that integrates advanced numerical
techniques to simulate both saturated and unsaturated flow conditions. This model has been
coupled with different land surface and atmospheric models to better understand the interactions
between the subsurface, surface, and atmospheric processes (Kollet and Maxwell, 2006; Maxwell
et al., 2007, 2011, 2014b). Herein, we introduce a newly developed coupled land surface and
subsurface hydrology model, ParFlow-LIS/Noah-MP (PF-LIS/Noah-MP) and study its
effectiveness and usefulness for simulating land surface and subsurface hydrologic processes. We
encourage the readers to refer to Fadji et al (2024) for more information about the coupled system.
This paper has been under review at the time of writing this manuscript. Our primary objective is
to study the degree to which the coupled PF-LIS/Noah-MP model (Fadji et al 2024) can contribute
to better representation of surface and subsurface processes over UCRB. In particular, we study
the extent to which the land surface water flux estimates in the LIS/Noah-MP model are improved
by coupling it with the ParFlow groundwater model. For this purpose, we compared the coupled
PF-LIS/Noah-MP and LIS/Noah-MP model estimates of soil moisture, streamflow, water table
depth and terrestrial water storage with a suite of in-situ and satellite observations over the UCRB
in the United States.
The paper is organized as follows: first, we briefly describe the ParFlow and LIS/Noah-
MP model. Next, we discuss the coupling framework. In the results and discussion section of the
paper, we provide a comparison of the model simulations against observations and explore how
the coupled system could improve understanding of the land surface processes.



## 2. ParFlow

ParFlow (PARallel Flow) (Ashby and Falgout, 1996; Jones and Woodward, 2001; Kollet and Maxwell, 2006) is an integrated, parallel model platform that simultaneously solves variably saturated three-dimensional Richards' equation throughout the entire subsurface (Kollet and Maxwell, 2008). ParFlow does not separate the phreatic and vadose zones, it employs a unified solution by solving the compressible Richards' equation everywhere in the subsurface. This inclusive methodology allows to obtain a realistic representation of groundwater dynamics, shaped by the underlying geology and topography. In addition to its capability to simulate subsurface flow, ParFlow also tackles the complexities of overland flow and surface runoff. This is accomplished through a combination of continuity or Manning's equations, implemented in either kinematic or diffusive formats. By integrating these surface water flow components, ParFlow offers a fully integrated system that simultaneously solves the partial differential equations (PDEs) governing both surface water and subsurface flow (e.g. Kollet and Maxwell, 2006). Importantly, this integration is achieved in a globally implicit manner, ensuring the robust and efficient solution of these interconnected processes at each time step. The terrain following grid formulation in ParFlow is important for accurately representing topography (Maxwell, 2013). By solving the three-dimensional Richards' equation for variably saturated groundwater flow, the model simulates lateral groundwater flow and replicates the spatial and temporal variations of the water table. It is important to note that groundwater may take a longer time (for example compared to soil moisture) to reach a steady-state due to such a complicated subsurface configuration, which makes it a computationally intensive problem to solve (Maxwell et al., 2014a).

## 3. LIS

Since the LIS framework has already been extensively described in the original papers (Kumar et al., 2006; Peters-Lidard et al., 2007), here we only briefly review its main components and features. Land surface modeling within LIS relies on three key inputs: (1) initial conditions, describing the land surface's starting state; (2) boundary conditions, encompassing the atmospheric fluxes or 'forcings' (upper boundary condition) and soil fluxes or states (lower boundary condition); and (3) parameters, which represent the soil, vegetation, topography, and other land surface characteristics. Using these inputs, Land Surface Models (LSMs) available within LIS (e.g., Community Land Model (CLM), Noah-MP, Variable Infiltration Capacity (VIC), Mosaic



and Hydrology with Simple SIB (HySSIB)) solve the governing equations of the soil-vegetation-
snowpack medium, and estimate the surface fluxes (i.e., sensible and latent heat, ground heat,
surface and subsurface runoff, and evapotranspiration) and states (i.e., soil moisture and
temperature, snow water equivalent and depth). One of the significant features of LIS is its high-
performance land surface modeling and Data Assimilation (DA) infrastructure (Kumar et al.,
2008b). Its DA capability enables users to utilize a wide range of in-situ and satellite observations,
integrating them into various land surface models (those mentioned above) to enhance their
predictive skill while accounting for the different sources of uncertainty involved in different
layers of simulation. The DA embedded within LIS provides a possibility to perform probabilistic
simulations, which facilitate uncertainty characterization/quantification and help risk assessment
and effective decision making in the case of studying extreme hydrologic processes, such as floods
and droughts, among others.
In this study, we used the Noah-MP LSM (Niu et al., 2011) within LIS (LIS/Noah-MP).
In LIS/Noah-MP, groundwater storage changes are represented using a simplified bucket-type
linear reservoir approach. This method tracks variations in groundwater storage based on inflow,
known as recharge, and outflows, which include capillary rise and base flow. It is important to
note that this approach does not explicitly consider complex hydraulic properties such as hydraulic
conductivity, a parameter typically used in soil moisture modeling and, by extension, groundwater
recharge prediction (Li et al., 2021).

**4. ParFlow-LIS**
Here we describe how we coupled the ParFlow and LIS models. As we mentioned earlier,
in the coupled system (PF-LIS/Noah-MP), when the precipitation reaches the ground and
infiltrates the soil, LIS estimates the land surface processes (such as evaporation and transpiration)
and then calculates the net downward water flux which is later used as input to feed the ParFlow
model. It should be noted that the land surface model (LIS/Noah-MP) and groundwater model
(ParFlow) share the top four soil layers as the coupled soil zone where the two systems
communicate. ParFlow utilizes the Richards' equation to estimate the soil moisture in the coupled
zone and in the other soil layers down to the bottom layer. In the PF-LIS/Noah-MP system, in
addition to the top four soil layers with depth ranges from 0-0.1m, 0.1-0.4m, 0.4-1m, and 1-2m,
there are six additional layers, each with varying soil depths, ranging from 2-7m, 7-17m, 17-42m,





42-92m, 92-192m, to the bottom layer from 192-492m. By using saturation data generated by
ParFlow as one of its outputs and incorporating the soil layer porosity values, the LIS/Noah-MP
model calculates the soil moisture content ($\theta$). This 3D moisture data, derived from ParFlow,
replaces the 1D soil hydrology within the LIS/Noah-MP model, affecting the simulation of other
land surface processes by LIS/Noah-MP. This is a two-way coupling; at each time step, LIS/Noah-
MP computes evaporation, transpiration, snowmelt, and throughfall and passes these to ParFlow
and then ParFlow feeds back a new soil moisture field to LIS/Noah-MP. Figure 1 schematically
illustrates the soil column, with red and green boxes delineating the control volumes for LIS/Noah-
MP and ParFlow, respectively.  Where these two areas overlap (shown with yellow arrow) is the
coupled soil zone (top four soil layers). The initial soil moisture condition starts from the land
surface with $\theta_{residual}$ ($\theta$ can be any value depending on condition) and varies down to the water
table depth, where the soil becomes saturated ($\theta_{saturation}$). Above the water table, the pressure
head is negative, while below the water table in the saturated soil zone, it becomes positive.
ParFlow provides estimates of pressure head and soil saturation, which, along with soil-specific
storage and porosity, are used to calculate subsurface storage. Through ParFlow, we can estimate
groundwater storage and lateral flow, both of which significantly impact the land surface energy
and water flux estimates within the land surface model. By integrating ParFlow with LIS/Noah-
MP, we can accurately estimate the groundwater storage and account for subsurface lateral flow,
facilitating the communication between the land surface and subsurface hydrologic processes.

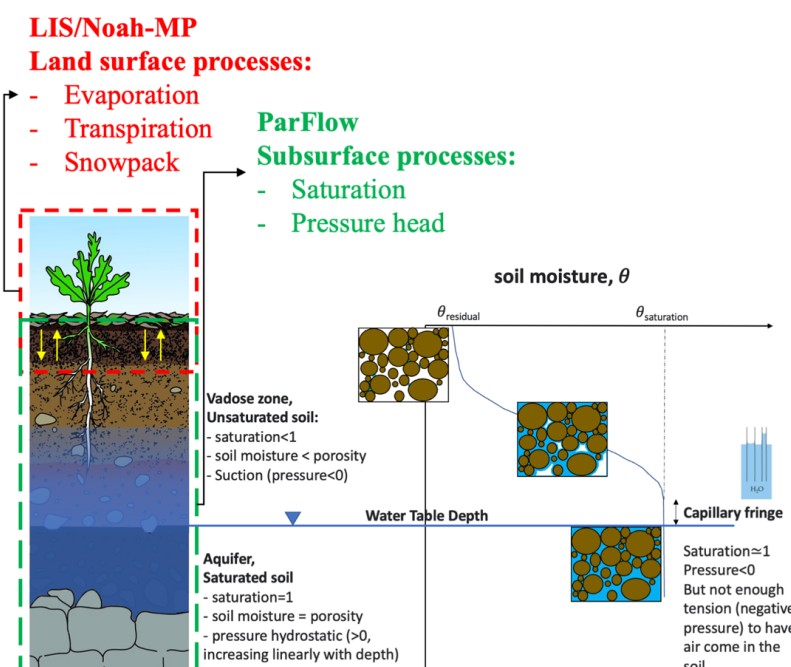

*Figure 1. Schematic of the coupled PF-LIS/Noah-MP model. Single soil column representing the coupling zone between the LIS/Noah-MP and ParFlow.*

## 5. Study Area

This study is conducted over the UCRB, a snow-dominated region covering approximately 280,000 km$^2$. Stretching from the river's origins in the Rocky Mountains of Colorado and Wyoming to its endpoint at Lee's Ferry in Northern Arizona, the basin exhibits a significant variation in elevation, ranging from 4,320 meters to 937 meters (Figure 2). Throughout the winter season, which encompasses the period from October through the end of April, the snow covered area within the UCRB fluctuates between 50,000 km$^2$ and 280,000 km$^2$. This seasonal change in snow covered area plays a pivotal role in both the energy dynamics and hydrological cycle of the region (Liu et al., 2015; Painter et al., 2012). The Colorado River is the primary water source for over 35 million people in the United States and an additional 3 million in Mexico. A recent publication by the US Geological Survey (Miller et al., 2016) indicates that up to half of the water coursing through the rivers and streams within the Upper Colorado River Basin originates from





groundwater sources. Recognizing the extent of available groundwater and understanding its
replenishment process holds significant importance for the sustainable management of both
groundwater and surface water resources within the Colorado River basin. For more information
about UCRB, its climatology and geology etc., we refer interested readers to Miller et al (2016).

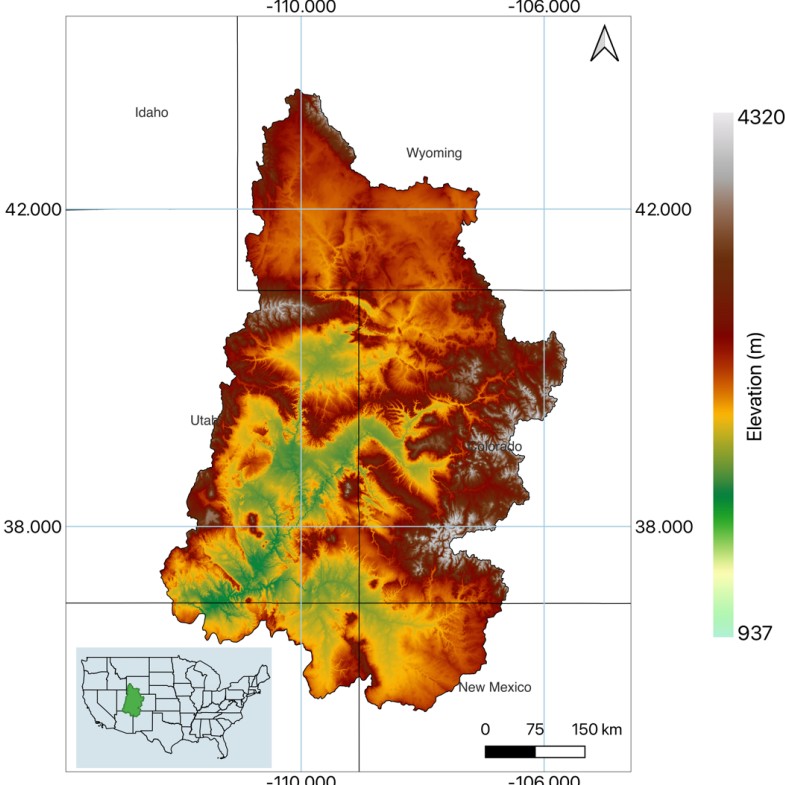


*Figure 2. Topography of the Upper Colorado River Basin (UCRB) and its location in the US.*

**6.  In-situ Observations and Satellite Products**

238        In this section we describe all those in-situ observations and satellite products that are used
for validation of model simulations. As for in-situ observations, we use soil moisture datasets
available from multiple observation networks over UCRB, USGS streamflow stations and
groundwater monitoring wells. The locations of these in-situ stations are shown in Figure 3. To
employ the maximum number of soil moisture stations covering the region, we used datasets
provided by ISMN (International Soil Moisture Network) which collected and compiled multiple





networks including, ARM (Atmospheric Radiation Measurement), PBO_H2O (Plate Boundary
Observatory), SCAN (Soil Climate Analysis Network), SNOTEL (SNOw TELemetry), USCRN
(U.S. Climate Reference Network), and iRON (Roaring Fork Observation Network). In total, we
have data from 238 soil moisture stations in the UCRB and its vicinity (see Figure 3). The
distribution of these stations by soil depth is as follows: Layer #1 (0-0.1 meters): 235 stations,
Layer #2 (0.1-0.4 meters): 218 stations, Layer #3 (0.4-1 meter): 216 stations, Layer #4 (1-2
meters): 41 stations. Having data from multiple depths improves the comparison with simulated
soil moisture and hence the evaluation of the coupled PF-LIS system. The soil moisture datasets
are publicly available at https://ismn.earth/en/. Streamflow and water table depth data are available
at https://waterdata.usgs.gov/nwis/rt and https://waterdata.usgs.gov/nwis/gw, respectively. We
made use of data from the period 2002 to 2022. In total, there are 374 UGSG stream stations and
18 USGS groundwater monitoring wells in the UCRB with observations from 2002 to 2022.
Measurements failing to meet the USGS quality control criteria (e.g., those flagged for potential
measurement inconsistency or negative outlier values) were removed.

258         In addition, we used two satellite products to investigate the effectiveness of PF-LIS/Noah-

MP in estimating the soil moisture and terrestrial water storage. For soil moisture, we use THySM
(Thermal Hydraulic disaggregation of Soil Moisture; Liu et al (2022)). This is a downscaled
version of SMAP (Soil Moisture Active Passive) satellite soil moisture data, which has 1-km
spatial resolution and is available on a daily time scale. THySM shows higher accuracy than the
SMAP / Sentinel-1 (SPL2SMAP_S) 1 km SM product when compared to in situ measurements.
Anomalies of Terrestrial Water Storage (TWS), derived from Gravity Recovery and Climate
Experiment (GRACE; Tapley et al., 2004) and GRACE Follow On (GRACE-FO; Landerer et al.,
2020) satellite observations, were compared to those from the coupled PF-LIS system. Launched
in 2002 and 2018, GRACE and GRACE-FO have provided monthly, global maps of fluctuations
in terrestrial water storage (i.e., the sum of groundwater, soil moisture, surface waters, snow and
ice), based on precise monitoring of variations in Earth's gravity field via its effects on the orbits
of a pair of twin satellites (http://www2.csr.utexas.edu/grace/RL05_mascons.html). The dataset
employed in this study, known as CSR Release-06 GRACE Mascon Solutions, was disseminated
by the Center for Space Research (CSR) at the University of Texas, Austin (Save et al., 2016). A
monthly TWS anomaly represents the current value minus the 2004 to 2010 mean. While GRACE
can detect TWS anomalies relative to the long term mean, it cannot quantify the absolute water





mass stored. Due to its relatively coarse spatial resolution (> 100,000 km$^2$) it has primarily been
used to study major river basins and other large regions (Rodell and Reager, 2023; Scanlon et al.,
2016). UCRB with approximately 280,000 km² area meets this criterion.

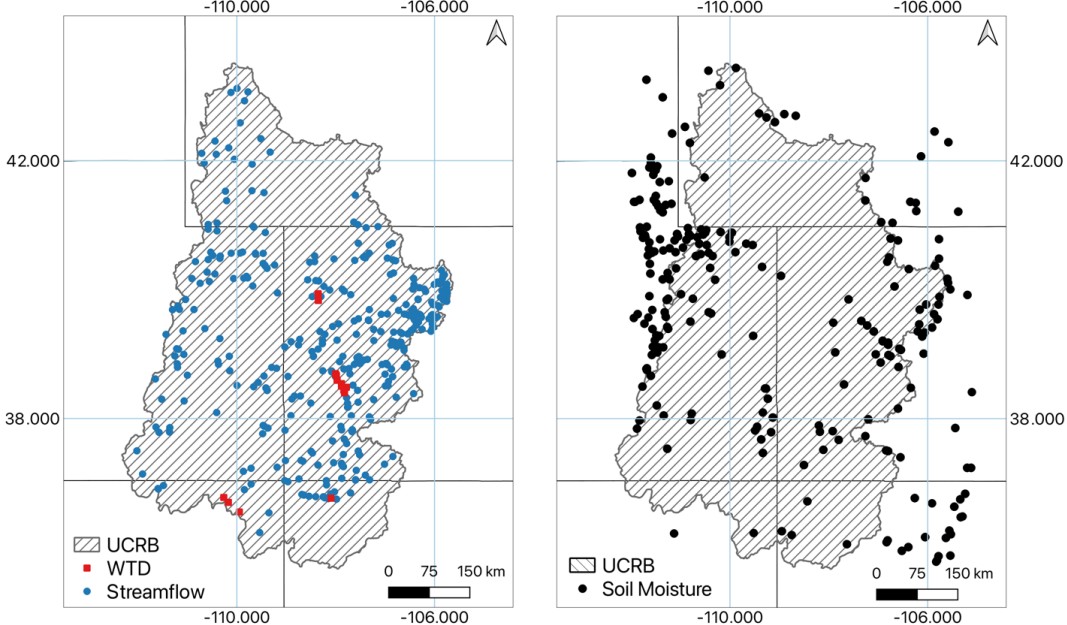


*Figure 3. Location of in-situ soil moisture, USGS streamflow, and WTD stations.*

## 7.  PF-LIS/Noah-MP Model Setup

### 7.1.  Input Datasets

In this study, we classified model parameters into two categories: surface and subsurface

characteristics. The surface parameters, which encompass topographic slopes and land cover data,
were determined as follows: Topographic slopes were calculated using the Priority Flow toolbox
(Condon and Maxwell, 2019), employing elevation data from the hydrological data and maps
derived from Shuttle Elevation Derivatives at multiple Scales (HydroSHEDS) as detailed and
tested in Zhang et al (2021). Land cover information was extracted from the National Land Cover
Database (NLCD) at a 30-meter resolution and subsequently rescaled to match the model's 1-
kilometer resolution (see Figure S1 in the supplementary file). The land cover values are based on



the classifications of the International Geosphere-Biosphere Program (IGBP). Regarding the
subsurface components of the ParFlow domain, they consist of four soil layers at the top (with
depths of 0.1, 0.3, 0.6, and 1 m, starting from the surface and totaling 2 m) and six geology layers
at the bottom (with depths of 5, 10, 25, 50, 100, and 200 m, starting from the surface and totaling
390 m). The development of the 3D subsurface, which includes soil, unconsolidated, a semi-
confining layer, bedrock aquifers, and the 3D model grid, is detailed in Tijerina-Kreuzer et al
(2024). The subsurface parameters (e.g. saturated hydraulic conductivity, porosity, and van
Genuchten parameters for the soil and subsurface) are detailed in Tijerina-Kreuzer et al (2024) and
Yang et al (2023). For the atmospheric forcing data, we use the phase-2 of the North American
Land Data Assimilation System (NLDAS-2) product (https://ldas.gsfc.nasa.gov/nldas/v2/forcing).
This dataset has eight variables: precipitation, air temperature, short-wave and long-wave
radiation, wind speed in two directions (east-west and south-north), atmospheric pressure, and
specific humidity.


**7.2. Model Spinup**

To be able to spinup the PF-LIS/Noah-MP model, we need to first spinup ParFlow and
LIS/Noah-MP individually and make sure both systems have the most realistic initial conditions.
The initial condition (i.e., pressure head) for the ParFlow model was directly obtained from Yang
et al (2023). who spunup the ParFlow model over the entire CONUS. We subsetted the UCRB
region from that initial pressure file. For more information about the ParFlow spinup process etc.
we refer the interested readers to Yang et al (2023). To spin up the LIS/Noah-MP model over
UCRB, we ran LIS/Noah-MP over 20 years (from 2002 to 2022) three times. To run the LIS/Noah-
MP model, we use the NASA Land surface Data Toolkit (Arsenault et al., 2018), to create the
LIS/Noah-MP domain file that encompasses all the parameters that LIS/Noah-MP requires to run.
Next, we use the initial conditions for both ParFlow and LIS/Noah-MP, to perform the PF-
LIS/Noah-MP model spinup. We ran the PF-LIS/Noah-MP over the period of water year 2005 (a
normal water year, not dry and not wet) six times, which was sufficient to bring the PF-LIS/Noah-
MP system into quasi-equilibrium.



## 8. Results and Discussion


In this section, we discuss the results of the PF-LIS/Noah-MP model simulations and aim to
gain a comprehensive understanding of how the coupled system can enhance the modeling of land
surface processes and provide an accurate representation of groundwater storage. Using the initial
conditions derived from the model's spinup process, we ran the PF-LIS/Noah-MP model over a
20-year period, spanning from 2002 to 2022. Concurrently, we ran the LIS/Noah-MP model for
the same time frame, facilitating a comparative analysis of the two model outputs. All model setup
and simulations were executed on the NASA Discover High-Performance Computing (HPC)
cluster. On average, a one-year simulation utilized approximately 295,000 core hours, resulting in
roughly one day of wall-clock time. The entire 20-year simulation consumed approximately 6
million core hours of computing time, extending over approximately 1.5 months of wall-clock
time.

### 8.1. Soil Moisture Analysis


Here, we study the extent to which the coupled system contributes to an improved
representation of soil moisture in the top four soil layers (referred to as the coupling soil zone),
where the two models interact. Figure 4 illustrates the topsoil moisture (with ~10 cm depth) as
simulated by the LIS/Noah-MP model (left panel) and the PF-LIS/Noah-MP model (right panel).
Note that the PF-LIS/Noah-MP simulations are limited to the UCRB region, which accounts for
the similarity in model results beyond the boundaries of this region. The results indicate that the
soil moisture output from the LIS/Noah-MP model generally aligns with the patterns of soil texture
and land cover. However, the soil moisture data generated by the PF-LIS/Noah-MP model
represents soil moisture distribution in a manner that closely correlates with topographical and
land surface characteristics. In a broad sense, both models demonstrate wet conditions across the
eastern UCRB and drier conditions towards the western regions. PF-LIS/Noah-MP provides soil
moisture data with higher spatial representativeness, which can be crucial for many applications.
For example, such finer spatial representations can be useful irrigation management applications,
which allows farmers to make better decisions about when and how much to irrigate, leading to
efficient water use and potentially higher crop yields.



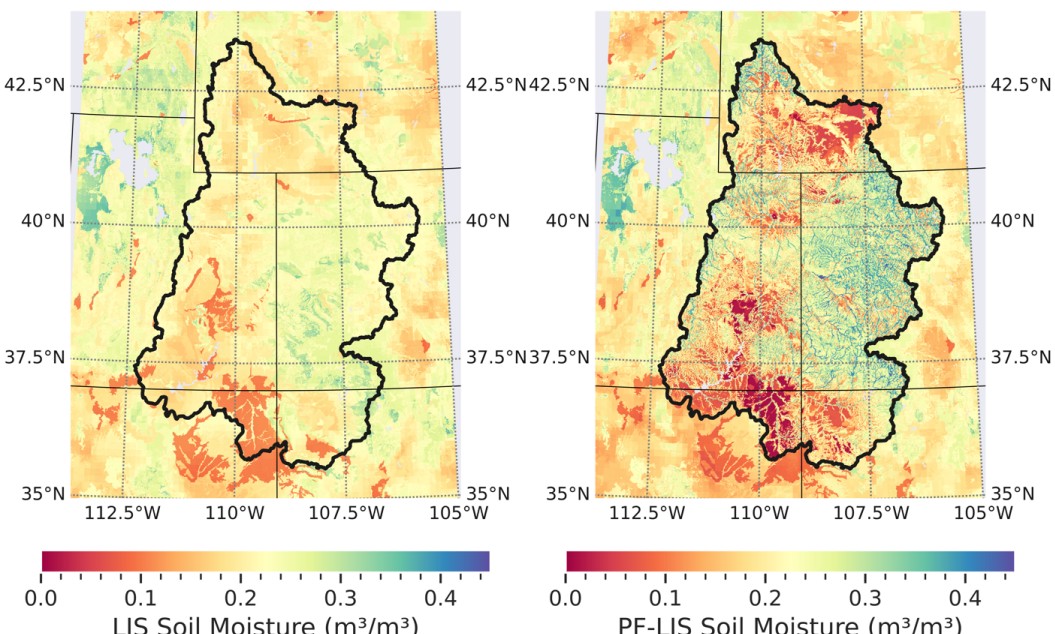


*Figure 4. Spatial pattern of topsoil layer moisture estimated by LIS/Noah-MP (left panel) and*
*PF-LIS/Noah-MP (right panel). This result is reported for 01/23/2002.*

To further study the model simulation results, we conducted a comparative analysis between PF-LIS/Noah-MP and LIS/Noah-MP-estimated soil moisture values and the satellite-based soil moisture product obtained from SMAP. As previously noted, our analysis employed downscaled soil moisture data with a spatial resolution of 1 kilometer, which is consistent with the resolution of the model simulation, thereby enhancing the accuracy of our comparative analysis. Figure 5 illustrates the outcomes, with the first row depicting the correlation coefficients and the second row showing the unbiased root mean square error (ubRMSE). The ubRMSE serves as a metric that SMAP utilizes for reporting product accuracy. The SMAP mission requirement for soil moisture product accuracy sets the ubRMSE at 0.040 m³/m³ (Chan et al., 2016). Due to the temporal coverage of the SMAP satellite, we calculated both performance metrics over the period of April 2015 to December 2022. To perform this, we used the NASA Land surface Verification Toolkit (LVT; Kumar et al. 2012), which enables rapid evaluation of model simulations by comparing against a comprehensive suite of in-situ, remote sensing, and model and reanalysis data products (https://lis.gsfc.nasa.gov). As shown in Figure 5, in general, both performance measures from both models show a similar spatial pattern across the UCRB. Further analysis revealed that,





particularly in regions characterized by higher altitudes and complex topography, PF-LIS/Noah-
MP-derived soil moisture values closely follow the SMAP observations, outperforming the
performance of LIS/Noah-MP-derived soil moisture.

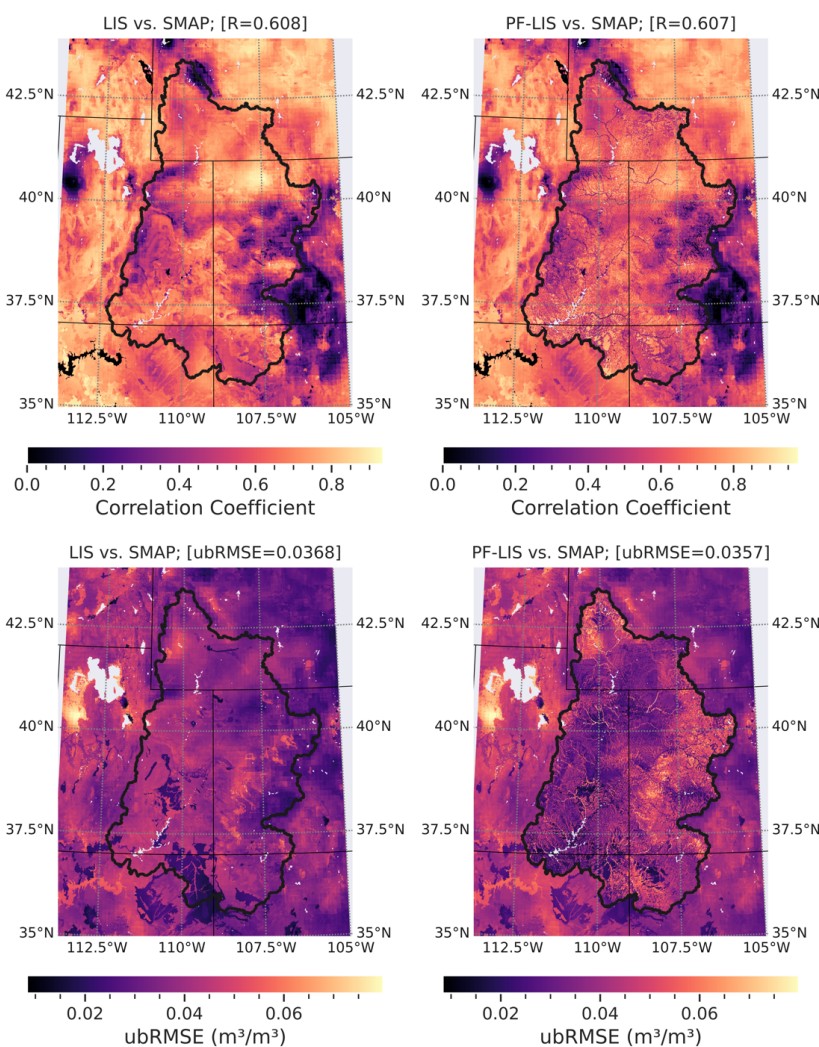

*Figure 5. The correlation coefficient and ubRMSE between the simulated topsoil*
*moisture and the SMAP product at 1-km spatial resolution. This result is reported for the period*
*of April 2015 to December 2022.*
The results also reveal that, in general, when we coupled ParFlow with LIS/Noah-MP, it
resulted in soil moisture fields with more spatial detail while keeping the accuracy in the same
range as compared to the LIS/Noah-MP standalone soil moisture estimates. ParFlow and





LIS/Noah-MP use a form of Richards' equation with some different assumptions. LIS/Noah-MP
uses a different function for retention (not the van Genuchten function used within ParFlow) and
it is 1D (one-dimensional).  The main difference between PF-LIS/Noah-MP and LIS/Noah-MP is
the deeper subsurface in PF-LIS/Noah-MP and the fact that it accounts for lateral flow, resulting
in a more physically realistic representation of water movement through the soil. This enables the
PF-LIS/Noah-MP model to capture the complex influence of topography and specific land surface
features on soil moisture.

389         Figure 6 illustrates the comparison between soil moisture estimates from the LIS/Noah-

MP and PF-LIS/Noah-MP models against in-situ networks in the UCRB and its adjacent regions.
In this section, we focus on presenting the comparison results for the topsoil (Figure 6) and root
zone (Figure 7) soil moisture, while the analysis for other soil depths can be found in the
supplementary file (Figures S2 and S3). The soil moisture comparison analysis was conducted
separately for each soil depth to study the effectiveness and utility of the coupled PF-LIS/Noah-
MP model in estimating soil moisture within the coupling soil zone. The 20-year simulation results
suggest that, across all four soil depths, the soil moisture values estimated by the PF-LIS/Noah-
MP model closely resemble those generated by the LIS/Noah-MP model. The regions' topography
(see Figure 2) and the results shown in Figure 5 collectively reveals that the coupled system
improves the accuracy of soil moisture estimates across the high altitudes with complex
topography in the UCRB. PF-LIS/Noah-MP utilizes the three-dimensional Richards' equation,
which is well-suited for accurately modeling soil moisture dynamics in regions with complex
topography due to its inherent features and mathematical formulation. The numerical solution of
the equation provides flexibility to handle complex boundary conditions in irregular terrains, while
its ability to incorporate spatial variability in hydraulic conductivity is vital for representing
changing soil properties across challenging landscapes. Moreover, it considers capillary rise and
gravitational effects, which are critical factors in areas with elevation changes. These attributes
collectively enable the PF-LIS/Noah-MP model to accurately simulate soil moisture dynamics in
regions characterized by complex topography. The results confirm that integrating the ParFlow
groundwater model with LIS/Noah-MP not only maintains the modeling performance of
LIS/Noah-MP but also enhances its ability to represent the spatial variability of land surface
processes, as previously demonstrated in Figures 4 and 5.

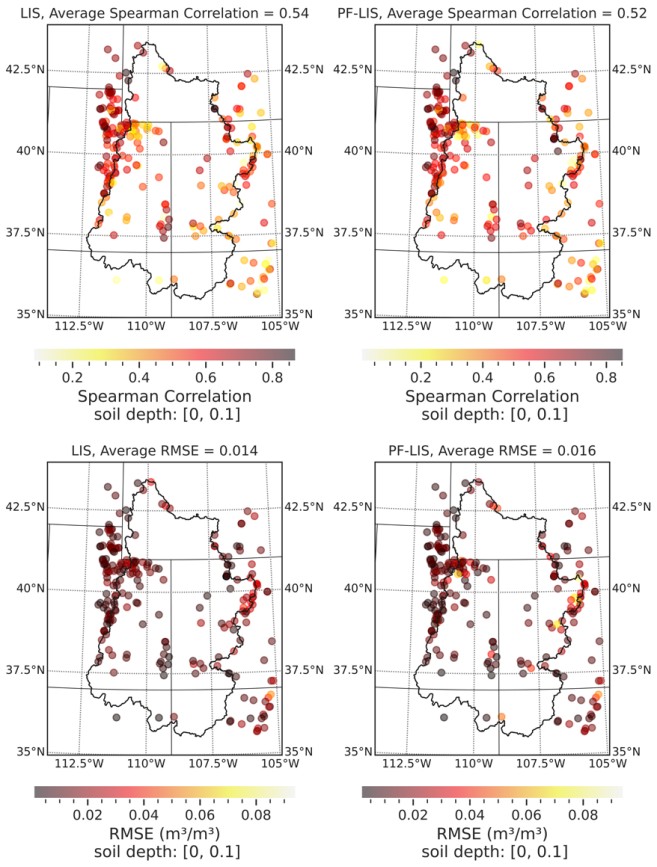


*Figure 6. The Spearman's correlation coefficient and RMSE between the simulated and observed soil moisture at the soil depth of 0-0.1 m. This result is reported based on 20-year model simulation and observation data, from January 2002 to December 2022.*


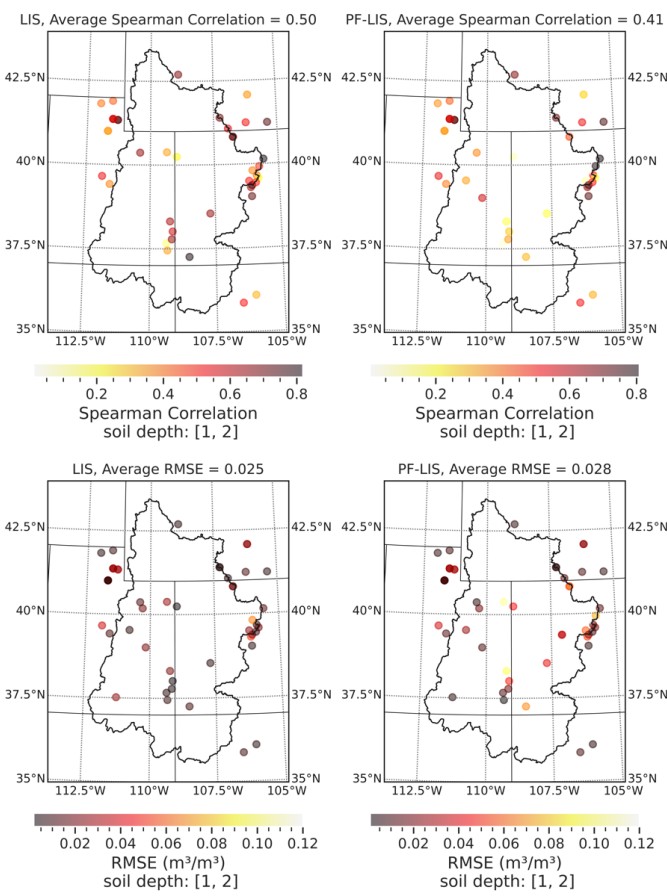

*Figure 7. The Spearman's correlation coefficient and RMSE between the simulated and observed*
*soil moisture at the soil depth of 1-2 m. This result is reported based on 20-year model*
*simulation and observation data, from January 2002 to December 2022.*

## 8.2. Streamflow Analysis

To calculate the streamflow at the location of the USGS stations, we used ParFlow hydrology module available on ParFlow GitHub page. For more information, we refer the interested readers to this page (https://github.com/parflow/parflow/tree/master/pftools). In particular, we used *calculate_overland_flow_grid* that requires different parameters to operate, these include pressure, slopex, slopey, mannings, grid size and the flow method (which is OverlandKinematic here). Figure S4 illustrates the total runoff over the study area for a certain day. We utilized two performance measures, namely Spearman's correlation (Rho) and Total Absolute Relative Bias, to assess the performance of our model on timeseries data. As explained



in Maxwell and Condon (2016), Tran et al., (2022), O'Neill et al., (2021) and Tijerina-Kreuzer et
al., (2021) plotting a graph (hereafter referred to as Condon Diagram) that visualizes these metrics
against each other provides a concise representation of the model's capability to accurately
simulate the timing and magnitude of streamflow. Spearman's Rho was employed to evaluate
disparities in timing between simulated and observed streamflow, while relative bias measured
differences in their volumes. A high Spearman's Rho value and a low relative bias value are
indications of when simulations closely match observations. If Spearman's Rho is less than 0.5 and
Total Absolute Relative Bias is less than 1, the model simulation produces accurate overall flow
estimates but does not match the hydrograph peaks well. Conversely, if Spearman's Rho is greater
than 0.5 and Total Absolute Relative Bias is less than 1, the model simulation is representing the
hydrograph shape (i.e. timing) with low flow bias. However, if Spearman's Rho is less than 0.5
and Total Absolute Relative Bias is greater than 1, the model simulation does not reproduce either
the flow magnitude or timing. On the other hand, if Spearman's Rho is greater than 0.5 and Total
Absolute Relative Bias is greater than 1, the model simulation represents the flow timing well but
not the overall flow magnitude. We excluded observations from stations influenced by human
activities (Falcone, 2011). While small drainage area basins may experience water withdrawals
and irrigation ditches, their susceptibility to anthropogenic influences is significantly lower
compared to larger drainage area basins, especially when considering monthly or annual scales
(Hao et al., 2008; Zhang et al., 2012). Therefore, we set a drainage area threshold of 500 km², and
stations with drainage areas exceeding this threshold underwent manual inspection. For example,
we removed the station at Lee's Ferry (drainage area: 289,560 km²), located just downstream of
the Glen Canyon Dam, from the analysis.

453       The left panel in Figure 8 shows the Condon Diagram, which summarizes the performance

of the PF-LIS/Noah-MP model in estimating streamflow across the USGS stations within the
UCRB region. The results indicate that the coupled system has reasonable skill in simulating the
streamflow. The right panel in this figure shows the spatial distribution of the USGS stations where
the model performance was evaluated. Figure S5 shows the simulated streamflow versus observed
streamflow over the period of 20 years at the monitoring location 9066510, which is associated
with a stream in Eagle County, Colorado (Spearman's Rho = 0.83 and RMSE=3.65 CMS). Overall,
the PF-LIS/Noah-MP model is able to adequately capture the magnitude and timing of streamflow
observations. This can be attributed to the robustness of the developed hydrology model, which
excels in precisely simulating base flow and its impact on overall streamflow. This lies in the
model's comprehensive integration of surface and subsurface hydrological processes. By
seamlessly incorporating both surface water and groundwater dynamics, the model achieves a level
of accuracy that allows it to effectively simulate streamflow time series, capturing the complex
interaction between the surface and subsurface physical processes. The low bias in model
simulations also indicates that the model is not systematically overestimating or underestimating
streamflow. This further suggests that the model's structure appears to be well-tailored to capture
the lateral and vertical water flow and its interaction with the land surface processes.

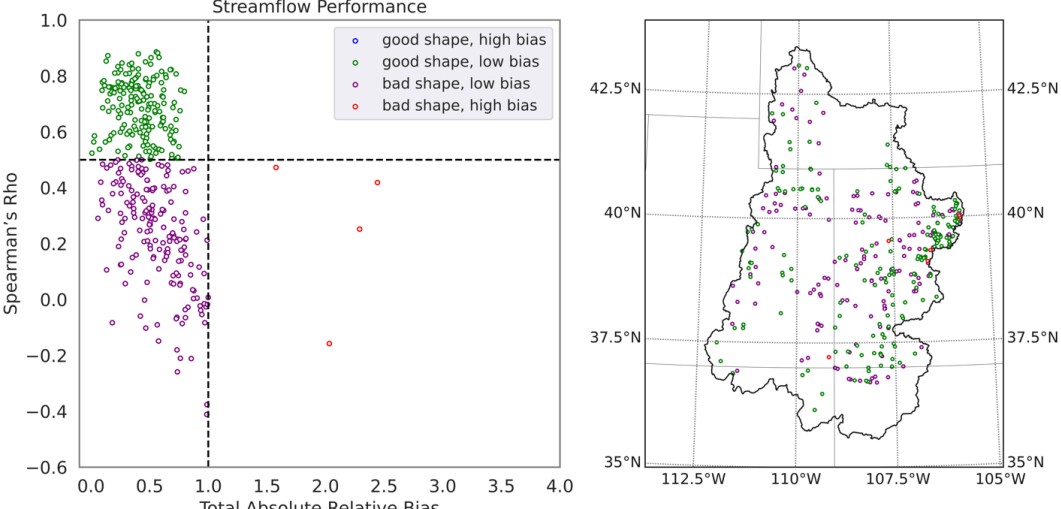

*Figure 8. Left panel: The Condon-diagram streamflow performance plot. Right panel: the*
*performance category of each gauge within the UCRB domain. This result is reported based on*
*20-year model simulation and observation data, from January 2002 to December 2022.*
**8.3.    Water Table Depth Analysis**
As mentioned earlier, the most important capability of the PF-LIS/Noah-MP model lies in
its ability to estimate groundwater levels up to 392 meters below the land surface. In this study,
we employed 10 soil layers with a cumulative depth of 392 meters. However, this depth can be
adjusted by the user based on the availability of geological information for the study region. Our
comparison of water table depth estimates from the PF-LIS/Noah-MP model with those observed
in USGS wells (refer to Table 1) reveals a general agreement between model simulations and
observations. However, in some locations the model performance is marginal due to the complex



topography of the UCRB. The higher bias observed can likely be attributed to the spatial resolution
of the PF-LIS/Noah-MP model. Deeper wells are typically located in mountainous regions
characterized by complex topography. It is important to note that all wells were assigned to the
nearest grid cell center without any additional adjustments. For example, the USGS station
382427107491401 is associated with a well in Montrose County, Colorado. This well, with a depth
of ~5 meters, is situated in close proximity to agricultural lands and central pivot systems
characterized by a predominantly flat topography. The dataset has been accessible since 2014, and
the reported values for Rho and bias stand at 0.65 and 0.34, respectively. However, at the USGS
station 395136108210000, linked to a well in Rio Blanco County, Colorado, with a depth of ~195
meters, located in a region characterized by more complex terrain and topography, the model's
performance is marginal. Water data has been accessible since 1975. Generally, the model's
performance is contingent upon the geographical locations of the stations. Stations located in
topographically complex surroundings tend to yield lower model performance compared to those
in areas with smoother and flatter environments. Some of the low skill values (reported in Table
1) could be a result of groundwater pumping impacts which are not represented within the
modeling framework.
*Table 1: Spearman correlation (Rho) and Total Absolute Relative Bias (TARB)*
*calculated between the water table depth estimated by PF-LIS/Noah-MP and observed by*
*USGS wells.*

| Rho | TARB | Latitude | Longitude | USGS Station ID |
|---|---|---|---|---|
| 0.196 | 0.98 | 36.490834 | -109.94817 | 362936109564101 |
| -0.79 | 0.98 | 36.647222 | -110.17068 | 363850110100801 |
| -0.29 | 0.81 | 36.715389 | -108.09297 | 364255108053202 |
| 0.62 | 0.97 | 36.727221 | -110.26319 | 364338110154601 |
| 0.65 | 0.34 | 38.4075 | -107.82056 | 382427107491401 |
| 0.59 | 0.08 | 38.448931 | -107.83547 | 382656107500701 |
| 0.63 | 0.20 | 38.488056 | -107.80861 | 382917107483101 |
| 0.54 | 0.20 | 38.496389 | -107.78278 | 382947107465801 |
| 0.06 | 0.48 | 38.514167 | -107.88194 | 383051107525501 |
| -0.32 | 0.77 | 38.554167 | -107.88111 | 383315107525201 |
| -0.28 | 0.41 | 38.607222 | -107.97083 | 383626107581501 |
| 0.75 | 0.19 | 38.685556 | -107.985 | 384110107591801 |
| -0.07 | 0.47 | 38.711111 | -108.00194 | 384240108000701 |
| -0.78 | 0.92 | 39.86 | -108.35111 | 395136108210000 |
| -0.25 | 0.94 | 39.86 | -108.35028 | 395136108210001 |
| -0.63 | 0.91 | 39.860133 | -108.35096 | 395136108210004 |





| 0.98 | 0.98 | 39.964444 | -108.35417 | 395755108211400 |
|------|------|-----------|------------|-----------------|
| 0.98 | 0.98 | 39.964722 | -108.35361 | 395755108211401 |


Figure 9, for example, illustrates the water table depth simulated by the PF-LIS/Noah-MP
model for a certain day over the UCRB. In general, our observations of water table depth maps
over UCRB show more deep water table depth in eastern areas with complex topography, such as
hilly or mountainous areas. These areas are often prone to localized variations in the water table.
However, regions with smoother topography, like plains, tend to have a more uniform water table
pattern, with gradual changes over larger distances. Human activities, such as drainage systems
and urbanization, can introduce variability in both types of environments. Overall, water table
dynamics are shaped by the interplay of topography, geology, and human influence, with complex
topography often contributing to more localized variations compared to smoother environments.

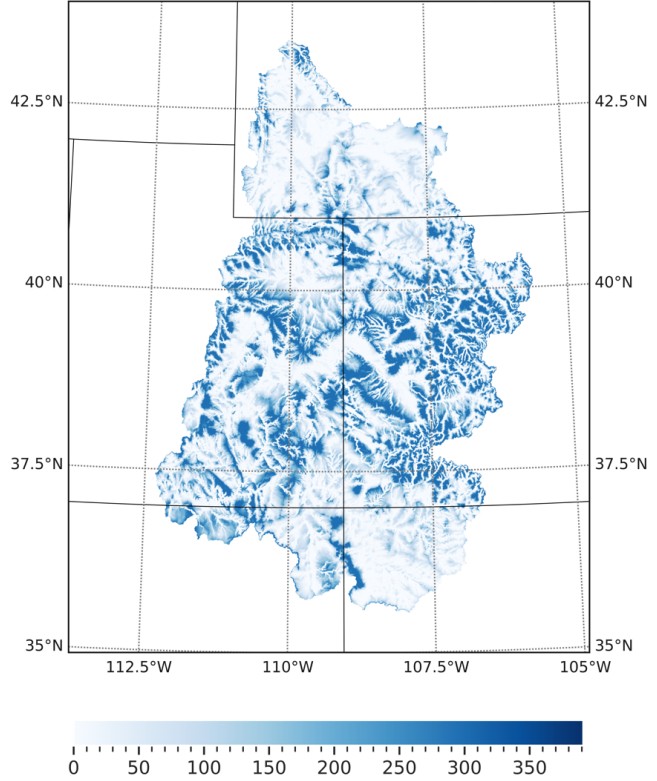

*Figure 9. Water table depth simulated by PF-LIS/Noah-MP model across the UCRB.*






## 8.4.    Terrestrial Water Storage Analysis

A comparison between changes in water storage from GRACE and GRACE-FO and the
PF-LIS/Noah-MP simulation for the period 2002 to 2022 is shown in Figure 10. The GRACE-
derived water storage anomalies were calculated by subtracting the mean water storage from 2004
to 2010. The same procedure was applied to the PF-LIS/Noah-MP outputs to maintain consistency
in the comparison. The two products demonstrated strong agreement throughout the period from
2002 to 2012, effectively capturing the drought years of 2003 and 2004, as well as the wet years
of 2005, 2008, and 2011. However, starting from 2013, there is a noticeable decline in the
agreement between the two time series, and this disparity becomes more pronounced during the
years 2020, 2021, and 2022. The observed disparity is likely attributed to the recent increased
anthropogenic effects on groundwater in the UCRB. The increased demand for water, driven by
population growth and agricultural expansion, has contributed to a decline in groundwater levels
(Carroll et al., 2024; Castle et al., 2014b; Miller et al., 2021; Tillman et al., 2022; Tran et al., 2022).
While this trend is accurately captured by the GRACE satellites, PF-LIS/Noah-MP underestimated
it. The integration of data assimilation into the coupled system can help to reconcile differences
between simulated and observed TWS. LIS already incorporates a data assimilation feature. In our
future work, we will study the extent to which the data assimilation capability embedded within
LIS improves the representation of the coupled system's response to TWS dynamics.

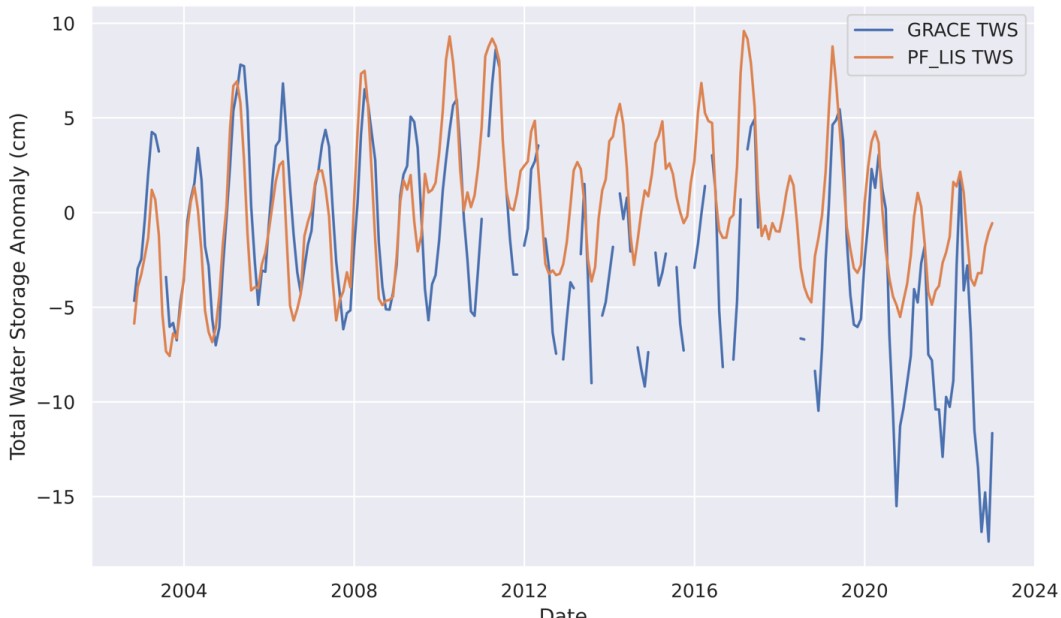

*Figure 10. Time series of the total water storage anomaly from the PF-LIS/Noah-MP model simulations and the GRACE and GRACE-FO observations.*

## 9. Conclusions

In this study, we introduced a coupled surface-subsurface hydrology model, PF-LIS/Noah-MP and studied its performance in estimating different hydrologic variables. This study was conducted in the UCRB, a region heavily dependent on groundwater to supply water for millions of people in the western United States. With an anticipated increase in drought occurrences due to climate warming, the region faces a heightened risk of groundwater depletion in the future. Understanding the dynamics of land surface and subsurface water in the UCRB is crucial for effective water resource management and policymaking. In this study, we employed the recently developed integrated surface-subsurface hydrology model, PF-LIS/Noah-MP, to assess key components such as soil moisture, streamflow, water table depth, and total water storage anomaly across the UCRB. These estimations were then compared with a comprehensive set of in-situ and satellite observations, encompassing soil moisture data from various networks, USGS streamflow and well observations, as well as satellite data from SMAP for soil moisture and GRACE for groundwater. The findings demonstrate that the integration of ParFlow with LIS/Noah-MP expands the physics represented by the LIS/Noah-MP model. These increased process





representations have two main advantages: better performance of land surface fluxes, especially in regions with complex topography, and accurate estimations of subsurface hydrologic processes, including water table depth. PF-LIS/Noah-MP presents a viable approach to studying land surface and subsurface hydrologic processes and their interactions across different scales. This research contributes valuable insights for informed decision-making in the management of water resources in the UCRB, particularly in the face of future climate challenges. The more detailed representation of subsurface processes within the PF-LIS/Noah-MP system also allows for improved utilization of remote sensing information through data assimilation. For example, to-date, the assimilation of GRACE terrestrial water storage observations has only been demonstrated within models that have a shallow groundwater representation and without the representation of lateral subsurface moisture transport processes (e.g., Kumar et al., 2016). The ongoing development will extend LIS' data assimilation capabilities to PF-LIS, to enable better exploitation of the information from remote sensing.

**Competing Interests**

At least one of the (co-)authors is a member of the editorial board of Hydrology and Earth System Sciences.

**Acknowledgments**

Financial support for this project was provided through NASA MAP 80NSSC20K1714. The simulations presented in this article were performed on NASA Discover cluster, provided by the NASA High-End Computing (HEC) Program through the NASA Center for Climate Simulation (NCCS). GRACE and GRACE-FO were jointly developed and operated by NASA, DLR and the GFZ German Research Centre for Geosciences.

**Authors Contributions**
P.A. wrote the first draft of the manuscript. P.A., F.M., C.Y., and D.R. created all the necessary files and datasets for model simulations. P.A. conducted the model simulations and validation analysis. P.A. and R.M. conceptualized the study. R.M., S.K., and M.R. edited the manuscript and helped with the model simulation analysis.





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

Supplemental Environmental Impact Statement for Near-term Colorado River Operations:
Tian, Y., Zheng, Y., Wu, B., Wu, X., Liu, J., and Zheng, C.: Modeling surface water-
groundwater interaction in arid and semi-arid regions with intensive agriculture, Environmental
Modelling and Software, 63, https://doi.org/10.1016/j.envsoft.2014.10.011, 2015.
Tijerina, D., Condon, L., FitzGerald, K., Dugger, A., O'Neill, M. M., Sampson, K., Gochis, D.,
and Maxwell, R.: Continental Hydrologic Intercomparison Project, Phase 1: A Large-Scale
Hydrologic Model Comparison Over the Continental United States, Water Resour Res, 57,
https://doi.org/10.1029/2020WR028931, 2021.
Tijerina-Kreuzer, D., Swilley, J. S., Tran, H. V., Zhang, J., West, B., Yang, C., Condon, L. E.,
and Maxwell, R. M.: Continental Scale Hydrostratigraphy: Basin-Scale Testing of Alternative
Data-Driven Approaches, Groundwater, 62, https://doi.org/10.1111/gwat.13357, 2024.
Tillman, F. D., Day, N. K., Miller, M. P., Miller, O. L., Rumsey, C. A., Wise, D. R., Longley, P.
C., and McDonnell, M. C.: A Review of Current Capabilities and Science Gaps in Water Supply
Data, Modeling, and Trends for Water Availability Assessments in the Upper Colorado River
Basin, https://doi.org/10.3390/w14233813, 2022.
Tran, H., Zhang, J., O'Neill, M. M., Ryken, A., Condon, L. E., and Maxwell, R. M.: A
hydrological simulation dataset of the Upper Colorado River Basin from 1983 to 2019, Sci Data,
9, https://doi.org/10.1038/s41597-022-01123-w, 2022.
U.S. Department of the Interior: Colorado River Basin SECURE Water Act Section 9503(c)
Report to Congress, 2021.
Wada, Y., Van Beek, L. P. H., Van Kempen, C. M., Reckman, J. W. T. M., Vasak, S., and
Bierkens, M. F. P.: Global depletion of groundwater resources, Geophys Res Lett, 37,
https://doi.org/10.1029/2010GL044571, 2010.
Wang, Y. and Chen, N.: Recent progress in coupled surface–ground water models and their
potential in watershed hydro-biogeochemical studies: A review, Watershed Ecology and the
Environment, 3, https://doi.org/10.1016/j.wsee.2021.04.001, 2021.
Williams, A. P., Cook, B. I., and Smerdon, J. E.: Rapid intensification of the emerging
southwestern North American megadrought in 2020–2021, Nat Clim Chang, 12,
https://doi.org/10.1038/s41558-022-01290-z, 2022.
Winter, T. C., Harvey, J. W., Franke, O. L., and Alley, W. M.: Ground Water and Surface Water
- A single Resource - U.S. Geological Survey Circular 1139, USGS Publications, Circular 1,
909   1998.

Xia, Y., Mitchell, K., Ek, M., Cosgrove, B., Sheffield, J., Luo, L., Alonge, C., Wei, H., Meng, J.,
Livneh, B., Duan, Q., and Lohmann, D.: Continental-scale water and energy flux analysis and
validation for North American Land Data Assimilation System project phase 2 (NLDAS-2): 2.
Validation of model-simulated streamflow, Journal of Geophysical Research Atmospheres, 117,
https://doi.org/10.1029/2011JD016051, 2012.





Yang, C., Tijerina-Kreuzer, D. T., Tran, H. V., Condon, L. E., and Maxwell, R. M.: A high-
resolution, 3D groundwater-surface water simulation of the contiguous US: Advances in the
integrated ParFlow CONUS 2.0 modeling platform, J Hydrol (Amst), 626,
https://doi.org/10.1016/j.jhydrol.2023.130294, 2023.
Yang, X., Hu, J., Ma, R., and Sun, Z.: Integrated Hydrologic Modelling of Groundwater-Surface
Water Interactions in Cold Regions, https://doi.org/10.3389/feart.2021.721009, 1 December
921  2021.

Zhang, A., Zhang, C., Fu, G., Wang, B., Bao, Z., and Zheng, H.: Assessments of Impacts of
Climate Change and Human Activities on Runoff with SWAT for the Huifa River Basin,
Northeast China, Water Resources Management, 26, https://doi.org/10.1007/s11269-012-0010-8,
925  2012.

Zhang, J., Condon, L. E., Tran, H., and Maxwell, R. M.: A national topographic dataset for
hydrological modeling over the contiguous United States, Earth Syst Sci Data, 13,
https://doi.org/10.5194/essd-13-3263-2021, 2021.