# Peer review of "Coupling the ParFlow Integrated Hydrology Model within the NASA Land 1 2 Information System: A case study over the Upper Colorado River Basin 3 1,4,7Peyman Abbaszadeh, 3Fadji Zaouna Maina, 1,4Chen Yang, 5Dan Rosen, 3Sujay Kumar, 4"

_Hydrology and Earth System Sciences, 2024_

## Referee Comment (RC2)

[referee-annotated manuscript omitted]

---

## Author Comment (AC1)

**Reviewer #1**

This manuscript explores a case study of a new model coupling, the intregrated hydrologic code ParFlow and the land surface code LIS/Noah-MP, to simulate hydrologic processes across the Upper Colorado River Basin (UCRB). The authors compare results between the standalone land surface model and the coupled model to in situ and remote sensing observations of soil moisture, streamflow and groundwater levels. When published, this paper will make a valuable contribution to the literature. At the moment, several aspects of the manuscript require clarification and revision before publication. Notably, the details of the model coupling are not fully described in this paper and are instead referenced in a manuscript currently under review, which limits my ability to evaluate the robustness of the methods and results. In addition, it the introduction and conclusions could be revised to clarify the novelty of the manuscript. Therefore, I am recommending major revisions.

We would like to thank the reviewer for their thoughtful comments and constructive suggestions. These insights have significantly enhanced both the quality and clarity of our work, helping us refine key aspects of the study and improve its overall readability. We appreciate the time and effort the reviewer dedicated to providing detailed feedback, which has been instrumental in strengthening the presentation and impact of our paper.

Major comments:

1) It is difficult to review this paper, as it presents a case study of a new model coupling, but the details of that coupling are described in a paper currently in review. Thus, I cannot evaluate the results presented here as the underling methods are not fully described. I recommend that the authors either (1) wait to publish this manuscript until the paper by Maida et al. (2024) is fully published, (2) publish a preprint of Maida et al. (2024), or (3) describe the ParFlow-LIS/Noah-MP coupling in depth.

This paper by Maina et al. (2025) was recently published in the Journal of Advances in Modeling Earth Systems. Here is the link to the paper: https://doi.org/10.1029/2024MS004415. We have updated the references in the revised manuscript.

*References*

*Maina, F. Z., Rosen, D., Abbaszadeh, P., Yang, C., Kumar, S. V., Rodell, M., & Maxwell, R. (2025). Integrating the interconnections between groundwater and land surface processes through the coupled NASA Land Information System and ParFlow environment. Journal of Advances in Modeling Earth Systems (JAMES), 17(2). https://doi.org/10.1029/2024MS004415*

2) I recommend the authors edit the introduction to emphasize the motivation for and novelty of this manuscript. ParFlow has long been coupled to one land surface model within the LIS framework (the Community Land Model), so what additional functionality is provided by coupling ParFlow to LIS? There are certainly advantages provided by the data assimilation and uncertainty estimation tools within LIS, but those tools are not used in this manuscript. Perhaps, then, the novelty of this paper is the difference in process representation between CLM and NoahMP, but the comparison to ParFlow-CLM is not presented in this paper.

ParFlow has not been previously coupled with any land surface model within LIS. This study, following our previously published work, is the first to explore the robustness of coupling ParFlow with Noah-MP within LIS to simulate land surface and subsurface hydrologic processes. We would also like to note that ParFlow is coupled to a different version of CLM than what is in LIS and that this version is incorporated into ParFlow.  That is it's not an external, community modeling platform.  So in addition to the differences between CLM and NoahMP, there are software differences too. To address this comment, we have revised the introduction section to emphasize the main novelty of this paper and its advantages.

*"The main novelty of this work is to demonstrate the capability of the newly coupled ParFlow and LIS/Noah-MP model in simulating land surface and subsurface hydrologic processes. Although LIS/Noah-MP has been widely used in many studies, its ability to model groundwater processes has been limited. In this study, we assess the performance of the ParFlow groundwater hydrology model when coupled with LIS/Noah-MP, focusing on its ability to simulate subsurface hydrologic processes, such as groundwater and soil water content, and their interactions with land surface processes. It is important to note that the primary goal of this paper is not to compare the performance of the ParFlow-LIS/Noah-MP system to LIS/Noah-MP or any other coupled system. Instead, the focus is on how ParFlow is integrated with LIS/Noah-MP and the resulting improvements, not only in simulating soil moisture (as accurately as LIS/Noah-MP) but also in enabling the simulation of groundwater and other subsurface*

*hydrologic processes, such as pressure head—processes that could not be modeled using LIS alone. Unlike LIS/Noah-MP, the ParFlow-LIS/Noah-MP coupling tracks subsurface water movement by solving the three-dimensional Richards equation, providing a more realistic representation of groundwater storage and water table dynamics."*

We have also updated figure 1 to better represent the main novelty of the proposed coupling system.

[Figure]

*Figure 1. Schematic of the coupled PF-LIS/Noah-MP model. Single soil column representing the coupling zone between the LIS/Noah-MP and ParFlow. θ$_{wp}$ and θ$_{fc}$ are wilting point and field capacity, respectively.*

3) While reviewing the results, I'm not sure I come to the same conclusions as the authors. The abstract (lines 33-34) and conclusions (lines 556-557) find that coupling ParFlow to LIS/Noah-MP improves accurat in regions with complex topography. However, the metrics presented in figures 5, 6, 7, S2 and S3 show that root mean squared error and correlation coefficients are nearly identical between LIS/Noah-MP and ParFlow-LIS/Noah-MP. However, those metrics are averaged over the entire domain, but perhaps there's a difference when those metrics are averaged over areas with complex topography? Please clarify.

Yes, over the regions with complex topography (regions with high elevation gradient – shown in Figure 2), the ParFlow-LIS/Noah-MP resulted in relatively better model performance in terms of soil moisture simulation compared to standalone LIS/Noah-MP model. To further clarify this point and address the reviewer's comment, we have added two additional figures to the revised manuscript, revised Figure 1, and added the following text to the revised manuscript.

[Figure]

*Figure 2. Topography of the Upper Colorado River Basin (UCRB) and its location in the US. Regions 1 and 2 represent the areas with complex topography - regions with high elevation gradient.*

*"In general, the results indicate that the coupled ParFlow-LIS/Noah-MP model produces soil moisture simulations comparable to those of the LIS/Noah-MP model across the entire UCRB. Figures 5, 8, and 9 (S2 and S3) show that the root mean squared error and correlation coefficient are nearly identical between the two models. For instance, in Figure 5, these metrics are reported as 0.036 m³/m³ and 0.608, respectively. However, further analysis—when these metrics are averaged over areas with complex topography—revealed that, in regions with a high elevation gradient (for instance, regions 1 and 2 shown in Figure 2), the ParFlow-LIS/Noah-MP model outperforms the standalone LIS/Noah-MP model in terms of soil moisture simulation. Figures 6 and 7*

*demonstrate the performance of the LIS/Noah-MP and PF-LIS/Noah-MP models compared to SMAP observations, specifically zooming in on two regions with latitude and longitude ranges: Region 1 (37°N to 38.2°N, -108°W to -106°W) and Region 2 (40.5°N to 41°N, -111°W to -109.5°W). In Region 1, the LIS/Noah-MP model yielded a ubRMSE of 0.0323 m³/m³ and a correlation coefficient (R) of 0.308, whereas the ParFlow-LIS/Noah-MP model showed slightly higher values of 0.0358 m³/m³ and 0.343, respectively. In Region 2, the LIS/Noah-MP model reported a ubRMSE of 0.0388 m³/m³ and R of 0.482, while the ParFlow-LIS/Noah-MP model performed better with a lower ubRMSE of 0.0330 m³/m³ and a higher R of 0.539. These regions were selected due to their complex topography characterized by high elevation gradients (see Figure 2)."*

[Figure]

*Figure 6. Spatial distribution of soil moisture performance metrics (ubRMSE and R) for Region 1 (shown in Figure 2), comparing LIS/Noah-MP and ParFlow-LIS/Noah-MP models against SMAP observations.*

[Figure]

*Figure 7. Spatial distribution of soil moisture performance metrics (ubRMSE and R) for Region 2 (shown in Figure 2), comparing LIS/Noah-MP and ParFlow-LIS/Noah-MP models against SMAP observations.*

Minor comments:

Lines 61-78: This paragraph focuses on the importance of simulating irrigation, groundwater pumping and other water management infrastructure, but those processes are not included in the simulations in this paper. Thus, it would be helpful to clarify why this paragraph is included here

Toward the end of the manuscript, we emphasize the significance of human impacts, particularly groundwater regulation, in the UCRB. This is connected to the discrepancies observed between the ParFlow-LIS/Noah-MP groundwater simulations and GRACE groundwater observations. Including this information in the

introduction helps highlight the role of groundwater pumping and irrigation in the UCRB, providing context for their influence on model validation. To further address this comment, we have added the following text to the revised manuscript.

*"While the current study does not directly simulate irrigation or groundwater pumping, these processes are critical in understanding the water dynamics of the region and can influence model outputs. The discrepancy between the ParFlow-LIS/Noah-MP groundwater simulations and GRACE groundwater observations, discussed later in the manuscript, highlights the importance of including such human impacts in hydrologic models. Although addressing these processes is not the primary objective of this study, this work serves as a foundational step toward that goal. As an alternative to direct simulation, data assimilation techniques could be employed in future research to incorporate observed groundwater data or other relevant measurements. This would enable better representation of irrigation and groundwater pumping processes in the model, improving simulation accuracy and addressing the observed mismatch in groundwater observations."*

Lines 105-108: The reference to WRF feels a little out-of-place since it isn't used in this study. It would be helpful to briefly introduce Noah-MP here instead, since it is mentioned in the abstract and in the next paragraph

We have removed the reference to WRF in this paragraph and have instead provided a brief introduction to Noah-MP. The following text has been added to the revised manuscript:

*"One of the land surface models available within LIS is Noah-MP (Niu et al., 2011), an advanced version of the Noah land surface model. Noah-MP is specifically designed to simulate a range of land surface processes, including soil moisture, temperature, snowpack dynamics, vegetation dynamics, and energy fluxes between the land surface and the atmosphere. It incorporates multiple soil layers, a detailed representation of vegetation types and their properties, and advanced treatments of surface energy exchanges, all of which are important for capturing the complexity of land-atmosphere interactions. In this study, we utilize the LIS framework with the Noah-MP model to simulate these land surface processes, which are critical for accurately representing hydrologic fluxes in the UCRB. Noah-MP's flexibility in representing diverse land surface characteristics allows for a more realistic simulation of hydrological processes such as evapotranspiration, infiltration, and runoff. Its detailed soil-vegetation-atmosphere interactions make it especially useful for understanding water fluxes in regions like the UCRB, where land surface conditions have significant impacts on groundwater recharge and surface water availability."*

Line 116: Should Fadji et al. (2024) be Maida et al. (2024)?

Corrected.

*"Maina et al., (2024)."*

Line 111: It would be helpful to mention some examples of these couplings (ParFlow-CLM, ParFlow-WRF, etc)

To address this comment, we have added the following text to the revised manuscript:

*"ParFlow is a robust and versatile groundwater model that integrates advanced numerical techniques to simulate both saturated and unsaturated flow conditions. This model has been coupled with different land surface and atmospheric models, such as the CLM (Community Land Model) and WRF (Weather Research and Forecasting model), to better understand the interactions between the subsurface, surface, and atmospheric processes*

*Some examples of ParFlow-CLM applications include studies by O'Neill et al. (2021), Tijerina et al. (2021), and Tijerina-Kreuzer et al. (2023), which highlight its use in high-resolution, coupled hydrology–land surface modeling at continental scales. O'Neill et al. (2021) introduced the ParFlow–CLM model (PFCONUSv1) configured over the U.S. to evaluate water balance components, identifying areas for model improvement, such as streamflow biases and shallow water table depth. Tijerina et al. (2021) compared two continental-scale, high-resolution models—ParFlow-CONUS v1.0 and WRF-Hydro—in the first phase of the Continental Hydrologic Intercomparison Project (CHIP), highlighting the importance of model performance evaluation in large-scale hydrologic predictions. Tijerina-Kreuzer et al. (2023) focused on the evaluation of subsurface property configurations for integrated hydrological modeling, emphasizing the significance of accurate datasets for effective model performance and recommending a 1 km resolution subsurface dataset for large-scale hydrologic modeling. All these studies are based on the ParFlow-CLM framework, underscoring its capability in simulating complex hydrological processes at continental scales.*

*Some examples of ParFlow-WRF applications include studies by Maxwell et al. (2011) and Xu et al. (2022), which highlight its use in coupled atmospheric and hydrologic modeling. Maxwell et al. (2011) introduced the PF-WRF model, coupling the WRF atmospheric model with ParFlow to simulate subsurface flow and overland flow. Their study, applied to the Little Washita watershed, demonstrated improvements*

*in water resources and wind-energy forecasting, particularly in simulating rainfall, runoff, and the effects of soil moisture on wind power output. Xu et al. (2022) used an integrated process model (IPM) combining WRF with ParFlow-CLM to simulate hydrometeorological conditions in the East River Watershed. Their findings highlighted the significant impact of subgrid-scale physics configurations on simulated hydrological metrics like discharge, snowpack, and evapotranspiration, providing guidance for future modeling in mountainous watersheds. Both studies showcase the versatility of ParFlow-WRF in simulating complex hydrologic processes."*

*References:*

*O'Neill, M. M. F., Tijerina, D. T., Condon, L. E., and Maxwell, R. M.: Assessment of the ParFlow–CLM CONUS 1.0 integrated hydrologic model: evaluation of hyper-resolution water balance components across the contiguous United States, Geosci. Model Dev., 14, 7223-7254, 10.5194/gmd-14-7223-2021, 2021.*

*Tijerina, D., Condon, L., FitzGerald, K., Dugger, A., O'Neill, M. M., Sampson, K., Gochis, D., and Maxwell, R.: Continental Hydrologic Intercomparison Project, Phase 1: A Large-Scale Hydrologic Model Comparison Over the Continental United States, Water Resour Res, 57, e2020WR028931, https://doi.org/10.1029/2020WR028931, 2021.*

*Tijerina-Kreuzer, D., Swilley, J. S., Tran, H. V., Zhang, J., West, B., Yang, C., Condon, L. E., and Maxwell, R. M.: Continental scale hydrostratigraphy: basin-scale testing of alternative data-driven approaches, Groundwater, n/a, https://doi.org/10.1111/gwat.13357, 2023.*

*Maxwell, R.M.; Lundquist, J.K.; Mirocha, J.D.; Smith, S.G.; Woodward, C.S.; Tompson, A.F.B. Development of a Coupled Groundwater–Atmosphere Model. Mon. Weather Rev. 2011, 1, 96–116.*

*Xu, Z., Siirila-Woodburn, E. R., Rhoades, A. M., and Feldman, D.: Sensitivities of subgrid-scale physics schemes, meteorological forcing, and topographic radiation in atmosphere-through-bedrock integrated process models: A case study in the Upper Colorado River Basin, EGUsphere [preprint], https://doi.org/10.5194/egusphere-2022-437, 2022.*

Lines 131-150: Description of ParFlow could be more clear

To address this comment, we have added an appendix to the revised manuscript.

*"Appendix*

*The ParFlow model operates in three distinct modes: (1) variably saturated; (2) steady-state saturated; and (3) integrated watershed flows. This adaptability enhances its utility across a range of hydrological scenarios. Here we summarize each mode following the work of Kollet and Maxwell (2006).*

***Variably Saturated Flow***

*ParFlow can operate in variably saturated mode through the well-known mixed form of the Richards' equation:*

$$S_s S_w(p) \frac{\partial p}{\partial t} + \phi \frac{\partial \left(S_w(p)\right)}{\partial t} = \nabla q + q_s \qquad (1)$$
$$q = - k_s k_r(p) \nabla(p - z) \qquad (2)$$

*where $S_s$ is the specific storage coefficient [L-1], $S_w$ is the relative saturation [–] as a function of pressure head $p$, $t$ is time, $\phi$ is the porosity of the medium [–], $q$ is the specific volumetric (Darcy) flux [LT−1], $k_s$ is the saturated hydraulic conductivity tensor [LT−1], $k_r$ is the relative permeability [–], which is a function of the pressure head $p$, $q_s$ is the general source or sink term [T−1] (includes wells and surface fluxes, e.g., evaporation and transpiration). $z$ represents depth below the surface [L]. ParFlow has been utilized for numerical simulations, including the modeling of river–aquifer exchange involving both free-surface flow and subsurface flow. It has also demonstrated efficacy in addressing highly heterogeneous problems under variably saturated flow conditions. For the situations where the saturated conditions are predominant, the steady-state saturated mode in ParFlow becomes a valuable tool.*

***Steady-State Saturated Flow***

*The fully saturated groundwater flow equation is expressed as follows:*

$$\nabla q - q = 0 \qquad (3)$$
$$q = - k_s \nabla P \qquad (4)$$

*where $P$ represents the 3-D hydraulic head-potential [L]. ParFlow does include a direct solution option for the steady-state saturated flow that is distinct from the transient solver. When studying more sophisticated or complex processes, such as when simulating a fully coupled system is of interest (i.e., surface and subsurface flow), an overland flow boundary condition is employed.*

**Overland Flow**

*Surface water systems are interlinked with the subsurface system; this interaction plays a critical role for rivers. However, explicitly representing the connections between the two systems in numerical simulations is a difficult task. In ParFlow, overland flow is implemented as a two-dimensional kinematic wave equation approximation of the shallow water equations. The continuity equation for two-dimensional shallow overland flow is expressed as follows:*

$$\frac{\partial \psi_s}{\partial t} = \nabla\left(\upsilon\psi_s\right) + q_s \qquad (5)$$

*where $\upsilon$ is the depth-averaged velocity vector [LT−1] and $\psi_s$ is the surface ponding depth [L]. Ignoring the dynamic and diffusion terms results in the momentum equation, which is known as the kinematic wave approximation:*

$$S_{f,i} = S_{o,i} \qquad (6)$$

*The $S_{f,i}$ and $S_{o,i}$ represent the friction [−] and bed slopes (gravity forcing term) [−], respectively. $i$ indicates the $x$ and $y$ directions in the following equations. Therefore, Manning's equation can be used to build a flow depth-discharge relationship as follows:*

$$\upsilon_x = \frac{\sqrt{S_{f,x}}}{n}\psi_s^{2/3} \qquad (7)$$

$$\upsilon_y = \frac{\sqrt{S_{f,y}}}{n}\psi_s^{2/3} \qquad (8)$$

*where $n$ is the Manning roughness coefficient [TL−1/3]. The shallow overland flow formulation (Eq. 9) assumes the vertical averaging of flow depth and disregards any vertical change in momentum within the surface water column. To incorporate vertical flow (from the surface to the subsurface or vice versa), a formulation that couples the system of equations through a boundary condition at the land surface becomes essential. We can modify Equation (5) to include an exchange rate with the subsurface, $q_e$:*

$$\frac{\partial \psi_s}{\partial t} = \nabla\left(\upsilon\psi_s\right) + q_s + q_e \qquad (9)$$

*In ParFlow, the overland flow equations are directly coupled to the Richards' equation at the top boundary cell under saturated conditions. Conditions of pressure continuity (i.e., equal pressures at the ground surface for the subsurface and surface domains) and flux at the top cell of the boundary between the subsurface and surface systems are assigned. Setting pressure head in Equation (1) equal to the vertically averaged surface pressure, $\psi_s$:*

$p = \psi_s = \psi$     (10)

*and the flux, $q_e$, equal to the specified boundary conditions (for example, Neumann-type boundary conditions):*

$q_{BC} = - k_s k_r \nabla(\psi - z)$     (11)

*and one solves for the flux term in Equation (10), the result becomes:*

$q_e = \frac{\partial \|\psi, 0\|}{\partial t} - \nabla \upsilon \|\psi, 0\| - q_s$     (12)

*where the $\|\psi, 0\|$ operator is defined as the greater of the quantities, $\psi$, 0. Putting the equations (10) and (11) together results in the following relationship:*

$- k_s k_r \nabla(\psi - z) = \frac{\partial \|\psi, 0\|}{\partial t} - \nabla \upsilon \|\psi, 0\| - q_s$     (13)

*As we see here the surface water equations are represented as a boundary condition to the Richards' equation. For more information about the coupled surface and subsurface flow systems in ParFlow, we refer the interested readers to Kollet and Maxwell (2006)."*

*References:*

*Kollet, S. J. and Maxwell, R. M.: Integrated surface-groundwater flow modeling: A free-surface overland flow boundary condition in a parallel groundwater flow model, Adv Water Resour, 29, https://doi.org/10.1016/j.advwatres.2005.08.006, 2006.*

Lines 148-149: "groundwater may take a longer time (for example compared to soil moisture) to reach a steady-state due to such a complicated subsurface configuration" Does "long time" here refer to simulation time or computational time? I'm unclear if this sentence is meant to describe the long time it takes to spin-up water content in the deep vadose zone due to slow rates of groundwater recharge, or if it refers to long computational time due to the difficulty of solving the Richards equation across a thicker vadose zone.

We thank the reviewer for this comment.  What we mean is really the former of the two; that groundwater and the deeper vadose zone takes a longer time to reach steady state than the shallower subsurface stores.  This longer simulation time can result in longer simulation times, but not due to difficulty in solving Richards' equation, just because the equilibrium times for the deep vadose zone are so long.  We have revised this sentence to read:

*"It is important to note that groundwater and the deeper vadose zone may take long simulation times (for example compared to shallow soil moisture) to reach a*

*steady-state due to slow rates of groundwater recharge and subsurface heterogeneity, which can make it a computationally expensive problem to solve (Maxwell et al., 2014)."*

*References:*

*Maxwell, R. M., Putti, M., Meyerhoff, S., Delfs, J. O., Ferguson, I. M., Ivanov, V., Kim, J., Kolditz, O., Kollet, S. J., Kumar, M., Lopez, S., Niu, J., Paniconi, C., Park, Y. J., Phanikumar, M. S., Shen, C., Sudicky, E. A., and Sulis, M.: Surface-subsurface model intercomparison: A first set of benchmark results to diagnose integrated hydrology and feedbacks, Water Resour Res, 50, https://doi.org/10.1002/2013WR013725, 2014*

Line 155: Which variables are included in the initial conditions? Soil moisture, surface temperature, what else?

It includes the total volumetric soil moisture and liquid water volume, soil temperature, canopy intercepted water (ice and liquid), canopy temperature, ground surface temperature, snow water equivalent and snow depth. For more information, please see section 3.1.1.3.2 in https://land-da-workflow.readthedocs.io/en/release-public-v1.2.0/CustomizingTheWorkflow/Model.html.

We have also revised the text to further address this comment.

*"Land surface modeling within LIS relies on three key inputs: (1) initial conditions, describing the land surface's starting state (i.e., total volumetric soil moisture and liquid water volume, soil temperature, canopy intercepted water (ice and liquid), canopy temperature, ground surface temperature, snow water equivalent and snow depth)... "*

Lines 168-171: I recommend mentioning that the LIS data assimilation framework is not used in this study

We have included this in the revised manuscript. Thank you.

*"Please note that the LIS data assimilation framework is not used in this study."*

Lines 181-182: Is the coupling specific to NoahMP? Or could someone use this same code to use VIC or HySSIB instead of NoahMP?

The coupling is specific to Noah-MP.

Section 4: The description of this coupling could be more detailed. How is transpiration from the root zone handled? Is transpiration only from the top soil layer or does NoahMP draw water from deeper layers as well? Also, how is overland flow handled? In this description (and the image in Fig 1), it appears as though ParFlow only simulates subsurface flow.

To address this comment, we added the following text to the revised manuscript.

*"The LIS/Noah-MP model is designed to simulate the energy and water fluxes at the land surface, along with key state variables like ET and its components, snow-related variables (such as SWE and snow cover), and infiltration. It computes the surface energy balance by representing vegetation with a detailed canopy model, incorporating its dimensions, orientation, density, and radiometric properties. A two-stream radiation transfer scheme is employed to account for the complex interactions of solar radiation within the canopy. For snow processes, the model features a multi-layer snowpack, capable of storing liquid water and simulating melt and refreeze processes. It also includes a snow interception component, which models the loading and unloading of snow, sublimation, and other snow-related processes. The ET and infiltration values (which combine snowmelt and rainfall) produced by LIS/Noah-MP are passed on to ParFlow. ParFlow then calculates the surface, soil, and subsurface hydrodynamics, generating important hydrological outputs such as water table depth, groundwater storage (derived from pressure-head and saturation), soil moisture, and streamflow (Maina et al., 2025). In particular, transpiration is computed by LIS/Noah-MP using the soil moisture computed by ParFlow. Within LIS/Noah-MP, transpiration is computed using a Penman-Monteith based approach, where stomatal resistance (influenced by solar radiation, vapor pressure deficit, temperature, and soil moisture) controls canopy conductance. Actual transpiration is obtained by scaling potential transpiration with a soil moisture stress function, considering vegetation type, root distribution, and dynamic LAI."*

*References*

*Maina, F. Z., Rosen, D., Abbaszadeh, P., Yang, C., Kumar, S. V., Rodell, M., & Maxwell, R. (2025). Integrating the interconnections between groundwater and land surface processes through the coupled NASA Land Information System and ParFlow environment. Journal of Advances in Modeling Earth Systems (JAMES), 17(2). https://doi.org/10.1029/2024MS004415*

Line 254: Should be "USGS stream stations"

Corrected.

Line 255: I'm surprised at how few monitoring wells there are. Have the authors considered adding water level measurments from either the Colorado Water Conservation Board or the Utah ? From a cursory glance (https://dwr.state.co.us/Tools/GroundWater/WaterLevels), it seems like there are many water level measurements not included in the USGS database. Also, what are the screened intervals for each well? If a well screen extends across multiple model cells, how are modeled and observed values compared?

Thank you for your suggestion. We have utilized the 18 USGS stations available within the study region. We also studied the observational datasets from the Colorado Water Conservation Board, as you recommended. However, most of these datasets are not recorded at a daily time scale, and some fall outside the period of our study, which limits their use for model simulation validation. Additionally, we studied the distribution of the wells and found that they are spread across the model grid cells in such a way that each grid cell has only one USGS station available for use. We also reviewed the USGS documentation for each station but did not find any information about the screened intervals. The only related information available was the well depth, which we have added to Table 1 in the revised manuscript. To determine the screened intervals, we need well logs or well-completion reports. Since these are not available, we can estimate the screened intervals (using rule-of-thumb method), such as screening 20% of the well depth.

| USGS Station ID | Well Depth (m) |
|---|---|
| 362936109564101 | 259.9 |
| 363850110100801 | 407.4 |
| 364255108053202 | 18.6 |
| 364338110154601 | 264.4 |
| 382427107491401 | 4.9 |
| 382656107500701 | 7.5 |
| 382917107483101 | 4.5 |
| 382947107465801 | 5.9 |
| 383051107525501 | 5.5 |
| 383315107525201 | 10.4 |
| 383626107581501 | 6.1 |

| USGS Station ID | Well Depth (m) |
|---|---|
| 384110107591801 | 4.4 |
| 384240108000701 | 6.9 |
| 395136108210000 | 195 |
| 395136108210001 | 265.4 |
| 395136108210004 | 75.9 |
| 395755108211400 | 384 |
| 395755108211401 | 534.8 |

Fig. 3: I recommend clarifying that WTD corresponds to "water table depth". Also, do all of these monitoring wells truly represent the depth to the water table? Or do they represent groundwater head? It's unclear whether these wells are screened across the water table.

Yes, the monitoring wells represent the depth to the water level according to the USGS webpage. We included the following text in the revised manuscript to further clarify this.

*"In this study, the monitoring wells are used to measure the depth to the water table (WTD), not groundwater head."*

Line 283: The manuscript could be improved by expanding this section and including additional details on model set up, such as boundary conditions and the extent and discretization of the domain. What are the lateral boundary conditions for the PF-LIS model? I assume that cells outside the UCRB would be inactive, but results for those cells are shown in Fig 4, 5, etc. Similarly, what is the extent of the LIS/Noah-MP domain? What are the lateral boundary conditions?

ParFlow is run over the UCRB, with areas outside the defined region masked out and are inactive in the simulation. We showed the model result only using LIS-Noah-MP on both maps to highlight the difference between using the coupled ParFlow/LIS-Noah-MP and standalone LIS-Noah-MP model. For example, to what extent the coupled system is able to provide more detailed predictions of land surface process across different regions with different land surface characteristics. The boundary conditions for ParFlow are set as no-flow (Neumann conditions) along the lateral edges of the region, reflecting the natural limits where lateral flow into the model domain is negligible. Similarly, the bottom layer is also assigned a no-flow condition, as the model extends deep enough to reach a zone where

vertical flow is minimal. At the top of the domain, overland flow conditions are applied, corresponding to the land surface.

We have added the following text to the revised manuscript to further address this comment.

*"The total extent of the UCRB model is 608 km in the east–west (x) direction and 896 km in the south–north (y) direction, with a horizontal resolution of 1 km. The model depth is 392 m and consists of 10 layers with variable thicknesses of 200, 100, 50, 25, 10, 5, 1, 0.6, 0.3, and 0.1 m from bottom to top. ParFlow is run over the UCRB, with areas outside the defined region masked out. We present model results using only LIS-Noah-MP on both maps to highlight the difference between the coupled ParFlow/LIS-Noah-MP system and the standalone LIS-Noah-MP model. This comparison demonstrates the extent to which the coupled system provides more detailed predictions of land surface processes across regions with varying land surface characteristics. The boundary conditions for ParFlow are set to no-flow (Neumann conditions) along the lateral edges of the region, reflecting the natural limits where lateral flow into the model domain is negligible. Similarly, the bottom layer is assigned a no-flow condition, as the model extends deep enough to reach a zone where vertical flow is minimal. At the top of the domain, overland flow conditions are applied, corresponding to the land surface Maina et al., (2025)"*

Line 293: 1 km lateral resolution?

Yes. Revised.

Line 296: Are these depths or thicknesses?

It refers to the thickness. We have included this information in our response to the above comment.

Line 317: Were these three 20-year periods run sequentially? Also, how was 60 years determined to be an adequate spin-up period? Are there metrics to determine whether the system is at dynamic steady state?

Yes, the model was run sequentially, and 60 years of simulation were sufficient to ensure that the system reached quasi-equilibrium.

Lines 320-321: How different were the initial conditions across the shared portions of the PF-LIS/NoahMP domain? How were differences in the two soil moisture fields reconciled before starting the first coupled simulation?

Prior to running the coupled ParFlow-LIS/Noah-MP spin-up simulation, both models—ParFlow and LIS/Noah-MP—were spun up individually. When we compared the soil moisture simulations from both models, the results were very similar.

Line 323: What metrics were used to determine that the system was at quasi-equilibrium?

We considered the system to have reached quasi-equilibrium when the total storage change was less than 1% of the potential recharge.

Line 329: What size time step was used for input forcing and the output analysis for these simulations? Hourly meteorological forcing? Daily pressure/saturation output?

We used hourly meteorological forcing to run the model and employed daily output for analysis.

Line 349: What is the difference in input forcing that provides this finer spatial resolution in PF-LIS/Noah-MP than in LIS/Noah-MP alone? Weren't both codes run using the same lateral resolution?

PF-LIS/Noah-MP and LIS/Noah-MP use a form of Richards' equation with some different assumptions. LIS/Noah-MP uses a different function for retention (not the van Genuchten function used within ParFlow) and it is 1D (one-dimensional). The main difference between PF-LIS/Noah-MP and LIS/Noah-MP is the deeper subsurface in PF-LIS/Noah-MP and the fact that it accounts for lateral flow, resulting in a more physically realistic representation of water movement through the soil. This enables the PF-LIS/Noah-MP model to capture the complex influence of topography and specific land surface features on soil moisture.

Fig. 4: What are the values outside of the UCRB watershed boundary and why does the resolution appear to be lower beyond that boundary in PF-LIS? Are those cells identical between the two simulations?

Yes, the cells are identical between the two simulations. The outer boundary of the UCRB was simulated using LIS/Noah-MP only. The right panel shows the simulated values for the interior of the UCRB when ParFlow is activated, and the coupled system is used for soil moisture simulation. This figure highlights the contribution of the coupled system in providing more detailed information about soil moisture simulation and its relationship to the region's topographic characteristics.

Line 360: How does the vertical resolution of the simulations compare to the SMAP penetration depth?

The soil moisture simulation at the topsoil layer (10 cm depth) from both the ParFlow-LIS/Noah-MP and LIS/Noah-MP models was compared with SMAP soil moisture data.

Fig. 5: In the caption, it could be useful to clarify the depth interval for the simulated soil moisture values.

We have added this to the revised manuscript.

*"The comparison of soil moisture was made using data from the 10 cm soil depth."*

Line 384: Could the difference in overland flow between PF-LIS/Noah-MP and LIS/Noah-MP also contribute to the increased spatial heterogeneity observed in PF-LIS/Noah-MP simulations?

Yes, this is correct. Figure S4 in the supplementary information also shows the surface runoff (along with its spatial heterogeneity) simulated by the coupled system, PF-LIS/Noah-MP.

Lines 399-400: It's not immediately clear from the figures that PF-LIS/Noah-MP improves the accuracy of soil moisture in high altitude regions. Is there an alternate figure that more clearly shows this result?

Thank you for the suggestion. We have addressed this point earlier in our response to comment #3 under "Major Comments".

Line 455: Why do you think PF-LIS/Noah-MP is unable to capture the timing of runoff? Is this due to errors in hydraulic conductivity, which cause inaccurate estimates of the timing of the rainfall-runoff response?

Over some USGS stations, PF-LIS/Noah-MP has shown marginal efficiency in capturing the timing of runoff, and this is likely not solely due to errors in hydraulic conductivity. As discussed in Maxwell and Condon (2016), the algorithm used for topographic processing resulted in spatial inconsistencies between the modeled and actual stream networks. To address this, USGS gauges were mapped to the PF-LIS/Noah-MP grid using nearest-neighbor mapping and manual adjustments, ensuring the gauges were correctly placed on the appropriate ParFlow stream cells. These inconsistencies in stream network representation may contribute to inaccuracies in runoff timing, in addition to any potential errors in hydraulic conductivity.

To further address this comment, we have added the following text to the revised manuscript, where we discuss the Condon diagram.

*"Over some USGS stations, PF-LIS/Noah-MP has shown marginal efficiency in capturing the timing of runoff, and this is likely not solely due to errors in hydraulic conductivity. As discussed in Maxwell and Condon (2016), the algorithm used for topographic processing resulted in spatial inconsistencies between the modeled and actual stream networks. To address this, USGS gauges were mapped to the PF-LIS/Noah-MP grid using nearest-neighbor mapping and manual adjustments, ensuring the gauges were correctly placed on the appropriate ParFlow stream cells. These inconsistencies in stream network representation may contribute to inaccuracies in runoff timing, in addition to any potential errors in hydraulic conductivity.*

*Reference*

*Maxwell, R. M. and Condon, L. E.: Connections between groundwater flow and transpiration partitioning, Science, 353, 377–380, https://doi.org/10.1126/science.aaf7891, 2016."*

Line 471: A minor point, but it could be useful to add to this diagram the number of points that are in each quadrant/category.

Thanks for the suggestion. We added the following text to the revised figure caption.

*"This diagram includes 177 purple points, 197 green points, 4 red points, and no blue points."*

Line 492: How were groundwater heads compared between simulated and observed values? Given that some cells are up to 200 m thick and PF-LIS/Noah-MP reports a single pressure value per cell, do these calculations assume hydrostatic equilibrium within a given cell to calculate the exact water table depth within that cell? Similarly, for wells that have a long screen length and are screened entirely below the water table, the reported water level measurements integrate pressure across the length of the screen.

The reviewer is correct that we assume hydrostatic equilibrium within a cell to interpolate the exact water table depth.  We generally do try to compare predicted and observed heads in a way that honors the well construction, integrating over the screen if confined.   However, it's often difficult to determine this from the observation database and errors can occur.  We have added some clarifying text to the sentence a few above this section:

*"It is important to note that all wells were assigned to the nearest grid cell center without any additional adjustments, that water table depths are interpolated within grid cells assuming a hydrostatic equilibrium and that information regarding screen depth and well construction are used in the comparison when available."*

Lines 495-497: "Stations located in topographically complex surroundings tend to yield lower model performance compared to those in areas with smoother and flatter environments." Would it be possible to include a figure in the supplement to support this statement? It might be more clear to show this relationship in a map rather than in a table of latitude and longitude values.

We have added the following figure to the supplementary file as you suggested. Thank you.

[Figure]

*Figure S6. Spatial distribution of the estimated WTD and its comparison with well observations across various locations featuring different land characteristics.*

Lines 528-530: This is an interesting result! Does this discrepancy also suggest that PF-LIS/Noah-MP underestimates evapotranspiration because croplands in the simulations do not receive any groundwater-fed irrigation? Another option for future work would be to compare remote-sensing-based estimates of ET with estimates from both PF-LIS/Noah-MP and LIS/Noah-MP.

We did not focus on evapotranspiration in this study, so we cannot be certain that PF-LIS/Noah-MP underestimates evapotranspiration because simulated croplands don't include groundwater-fed irrigation. In our first paper (Maina et al., 2025), we compared evapotranspiration simulations from PF-LIS/Noah-MP and LIS/Noah-MP in irrigated areas. That paper gives more details about how the coupled system handles evapotranspiration, and we recommend the reviewer refer to it for more information.

Thank you for suggesting ideas for future work. There are many observational datasets and methods available to validate the PF-LIS/Noah-MP model outputs. In this paper, we focused on a subset of them, but future studies will expand the validation using additional observational data and covering other regions.

---

## Author Comment (AC2)

**Reviewer #2**

Abbaszadeh et al. present a model based study over the Upper Colorado River Basin. Using variety of in situ and satellite observations, a coupled model representing surface and subsurface hydrological processes (ParFlow/LIS-Noah-MP) is compared to a stand-alone LIS-Noah-MP implementation which includes much simpler subsurface representation. The main conclusion is that the coupled model enables more spatial detail to be captured in various states, but that statistical fit metrics with observations change little (or even slightly decline).

Overall the study is well written and illustrated, and I am in favour of its eventual publication in HESS. However, there are a few aspects which I think could be improved. For example:

Thank you for the positive feedback and super useful comments and suggestions. These insights have significantly enhanced both the quality and clarity of our work, helping us refine key aspects of the study and improve its overall readability. We appreciate the time and effort the reviewer dedicated to providing detailed feedback, which has been instrumental in strengthening the presentation and impact of our paper.

The abstract could benefit from being slightly more specific

We further revised the abstract and included more details.

"In general, the results show that the coupled ParFlow-LIS/Noah-MP model produces soil moisture simulations comparable to those of the LIS/Noah-MP model across the entire UCRB. The root mean squared error and correlation coefficients are nearly identical between the two models. However, further analysis—when these metrics are averaged over areas with complex topography—revealed that in regions with a high elevation gradient, the ParFlow-LIS/Noah-MP model outperforms the standalone LIS/Noah-MP model in terms of soil moisture simulation."

The assertions made in the Introduction (L94) that most previous large-scale subsurface modelling studies have not accounted for surface processes isn't really the case any more (as demonstrated by the extensive work on this topic by at least one of the co-authors).

Thank you for the suggestion. To address this comment we have decided to remove this sentence from the text.

Since ParFlow has previously coupled with various LSM and atmospheric models, the motivation for undertaking the specific coupling presented in this paper is not very clear. After reading, it seems that this may come down to the possibility to do Data Assimilation, which the LIS framework enables. However, this is not stated, and it is unclear that any DA was actually conducted.

ParFlow has not been previously coupled with any land surface model within LIS. This study, following our previously published work, is the first to explore the robustness of coupling ParFlow with Noah-MP within LIS to simulate land surface and subsurface hydrologic processes. We also would like to mention that in this study we have not done any data assimilation (DA). As you stated, this is one of the capabilities of the LIS framework, and we believe using the developed coupled system within LIS will enable assimilating satellite observations into PF-LIS/Noah-MP, improving its prediction skills while accounting for uncertainties.

To address this comment, we have revised the introduction section to emphasize the main novelty of this paper and its advantages.

"The main novelty of this work is to demonstrate the capability of the newly coupled ParFlow and LIS/Noah-MP model in simulating land surface and subsurface hydrologic processes. Although LIS/Noah-MP has been widely used in many studies, its ability to model groundwater processes has been limited. In this study, we assess the performance of the ParFlow groundwater hydrology model when coupled with LIS/Noah-MP, focusing on its ability to simulate subsurface hydrologic processes, such as groundwater and soil water content, and their interactions with land surface processes. It is important to note that the primary goal of this paper is not to compare the performance of the ParFlow-LIS/Noah-MP system to LIS/Noah-MP or any other coupled system. Instead, the focus is on how ParFlow is integrated with LIS/Noah-MP and the resulting improvements, not only in simulating soil moisture (as accurately as *LIS/Noah-MP)* but also in enabling the simulation of groundwater and other subsurface hydrologic processes, such as pressure head—processes that could not be modeled using LIS alone. Unlike LIS/Noah-MP, the ParFlow-LIS/Noah-MP coupling tracks subsurface water movement by solving the three-dimensional Richards equation, providing a more realistic representation of groundwater storage and water table dynamics."

The manuscript refers to a paper by Fadji et al. (2024) in which the coupled framework that is applied here is presented. However, it is not listed in the reference list? Is it available to reviewers (e.g. as a pre-print). Since the present paper depends on this

framework, this seems like a major omission. In this paper, there seems to be some confusion in the description of how the coupling is achieved.

This paper by Maina et al. (2025) was recently published in the Journal of Advances in Modeling Earth Systems. Here is the link to the paper: https://doi.org/10.1029/2024MS004415. We have updated the references in the revised manuscript.

**References**

Maina, F. Z., Rosen, D., Abbaszadeh, P., Yang, C., Kumar, S. V., Rodell, M., & Maxwell, R. (2025). Integrating the interconnections between groundwater and land surface processes through the coupled NASA Land Information System and ParFlow environment. Journal of Advances in Modeling Earth Systems (JAMES), 17(2). https://doi.org/10.1029/2024MS004415

Whilst I appreciate the supplemental file, I do not see any comment regarding the availability of the model configuration files and outputs. I strongly encourage the authors to make these materials available to the community so that other can build on their work, and to enhance the transparency and reproducibility of such advanced computational work.

We have added the "Data Availability Statement" section to the revised manuscript.

"Data Availability Statement: ParFlow-LIS is included in the Nasa Land Information System (LIS), an open-source software that can be found at Rosen and Dunlap (2024). NUOPC CAP has been integrated in both ParFlow and LIS. The data set and model configuration for LIS/Noah-MP and ParFlow-LIS/Noah-MP models can be found at Maina (2024)"

**References:**

Rosen, D., & Dunlap, R. (2024). fadjimaina/NASA-Land-Coupler: ParFlow-LIS (Version release\_0.1). Zenodo. https://doi.org/10.5281/zenodo. 14058196

Maina, F. (2024). Integrating the interconnections between groundwater and land surface processes through the coupled NASA Land Information System and ParFlow environment [Dataset]. Zenodo. https://doi.org/10.5281/zenodo.10950634

Please see the attached annotated PDF for my detailed comments.

Line 23: In my view, the syntax "that enable....to be studied" would be preferable.

We revised the sentence as you suggested.

"that enable the Earth's land surface and subsurface hydrologic processes to be studied."

Line 24: Consider changing to: "The integration... to harness their strengths provides an opportunity..."

We revised the sentence as you suggested.

"The integration of ParFlow and LIS/Noah-MP models to harness their strengths provides an opportunity to simulate surface terrestrial water processes and groundwater dynamics together, while enhancing the accuracy and scalability of hydrological modeling."

Line 25, 26: This statement is quite general and could perhaps be made more specific. How more precisely could (or does) this approach advance the state of the art or compare with what is typically done? For instance, since ParFlow considers coupled surface-subsurface processes, I am wondering at this stage what LIS/Noah-MP will contribute? I guess the key additional element is the atmosphere?

Thank you for the comment. To clarify, the ParFlow model is a subsurface hydrology model (groundwater model) and LIS/Noah-MP is a land surface model.

Line 29: I understood that ParFlow (and certainly ParFlow-CLM) does this already. So, are you essentially proposing/testing coupling to another LSM?

Yes. In this project, we coupled the ParFlow groundwater model with the LIS/Noah-MP land surface model, and the coupled system is basically a subsurface-surface hydrology model (similar to ParFlow-CLM as you mentioned).

Line 30, 31, 32: Good clear statement on the experimental design.

Thank you for the comment.

Line 34, 35: Consider slight language change: ", but also enables accurate simulation of subsurface hydrologic processes."

We revised the sentence as you suggested.

"This analysis confirmed that integrating ParFlow with LIS/Noah-MP not only enhances the capability of LIS/Noah-MP in estimating land surface processes over regions with complex topography but also enables accurate simulation of subsurface hydrologic processes."

Line 43: Insert "either" for greater clarity. Line 44: Is "be returned" better?

We revised the sentence as you suggested.

"For instance, precipitation that falls on the land surface can either infiltrate the soil and become soil moisture or runoff into nearby streams and rivers. Soil moisture can be returned to the atmosphere through."

Line 51: Change to: ", as well as their interactions..." Also, should feedback be plural?

We revised the sentence as you suggested.

"Climate change can impact surface and subsurface hydrologic processes, as well as their interactions and feedbacks to the atmosphere."

Line 55: Moreover? Or Furthermore?

We revised the sentence as you suggested.

*"Furthermore, human activities, such as irrigation and water pumping, can alter the natural behavior of surface–subsurface interaction..."*

Line 68: Consider making the specific link with irrigation here; groundwater is pumped and used in irrigation for agriculture.

We revised the sentence as you suggested.

"Groundwater pumping is an important source of water for irrigation in agriculture in the UCRB, particularly when and where surface water availability is limited." Line 69: I would argue that if pumping is excessive or unsustainable, it *does* lead to aquifer depletion. Same comment could apply above when you write "human activities can modify..." Line 71: Not only of agricultural. Isn't the point that the water is also then unavailable for other applications too?

We revised the sentence as you suggested.

*"Excessive pumping can lead to the depletion of aquifers, impacting water availability and the long-term sustainability of agricultural practices."*

Line 83: I would avoid the term "layers" here. Components? Facets?

We revised the sentence as you suggested.

"...affect different components of the hydrologic system..."

Line 85: Change to: "these connections" (agreement).

We revised the sentence as you suggested.

"...to better understand these connections has been..."

Line 94, 95: Not sure this is completely true anymore. Most of these citations are rather dated. For example, there's the work applying ParFlow across the CONUS and the coupled global model of de Graaf, etc. Subsurface processes have already been accounted for at a large scale. The question is whether the grid resolutions and subsurface (geological) data used in those studies are useful for local-scale water management applications.

Thank you for the suggestion. Totally agree. To address this comment we have decided to remove this sentence from the text.

Line 98: Check this.

Corrected.

Line 99: Please explain here the relationship between LIS and Noah-MP a little more. Right now, it comes out of nowhere. For instance, is the model really Noah-MP, but it is run within the framework of LIS, which enables integration with satellite observations?

Thanks for the suggestion. We have revised this part of the introduction to address this comment.

"The NASA Land Information System (LIS) is a software framework designed to facilitate the integration of land surface models and satellite remote sensing data for improved understanding and prediction of land surface processes. LIS is a modeling framework that offers a variety of model options. One of the key models that can be run within the LIS framework is the Noah-MP (Multi-Parameterization) model, which is a widely used land surface model. The LIS framework enables the coupling of the Noah-MP model with satellite observations and other data sources, providing a more comprehensive view of land-atmosphere interactions. Specifically, LIS serves as a tool for executing Noah-MP simulations, allowing for real-time integration of remote sensing data and enhancing the model's predictive capabilities."

Line 113, 114: Given the previous coupling efforts, what was the motivation for doing this one? For example, is LIS/Noah-MP expected to be stronger than previously coupled LSM?

ParFlow has not been previously coupled with any land surface model within LIS. This study, following our previously published work (Maina et al. 2025), is the first to explore the robustness of coupling ParFlow with Noah-MP within LIS to simulate land surface and subsurface hydrologic processes. We would also like to note that ParFlow is coupled to a different version of CLM than what is in LIS and that this version is incorporated into ParFlow. That is it's not an external, community modeling platform. So in addition to the differences between CLM and NoahMP, there are software differences too. To address this comment, we have revised the introduction section to emphasize the main novelty of this paper and its advantages.

"The main novelty of this work is to demonstrate the capability of the newly coupled ParFlow and LIS/Noah-MP model in simulating land surface and subsurface hydrologic processes. Although LIS/Noah-MP has been widely used in many studies, its ability to model groundwater processes has been limited. In this study, we assess the performance of the ParFlow groundwater hydrology model when coupled with LIS/Noah-MP, focusing on its ability to simulate subsurface hydrologic processes, such as groundwater and soil water content, and their interactions with land surface processes. It is important to note that the primary goal of this paper is not to compare the performance of the ParFlow-LIS/Noah-MP system to LIS/Noah-MP or any other coupled system. Instead, the focus is on how ParFlow is integrated with LIS/Noah-MP and the resulting improvements, not only in simulating soil moisture (as accurately as LIS/Noah-MP) but also in enabling the simulation of groundwater and other subsurface hydrologic processes, such as pressure head—processes that could not be modeled using LIS alone. Unlike LIS/Noah-MP, the ParFlow-LIS/Noah-MP coupling tracks subsurface water movement by solving the three-dimensional Richards equation, providing a more realistic representation of groundwater storage and water table dynamics."

References:

Maina, F. Z., Rosen, D., Abbaszadeh, P., Yang, C., Kumar, S. V., Rodell, M., & Maxwell, R. (2025). Integrating the interconnections between groundwater and land surface processes through the coupled NASA Land Information System and ParFlow environment. Journal of Advances in Modeling Earth Systems (JAMES), 17(2). https://doi.org/10.1029/2024MS004415

Line 117: Is it available to reviewers? It does not appear in the reference list!

This paper has recently been published. Please see the following reference. Therefore, we have removed that sentence "This paper has been under review at the time of writing this manuscript" from the text.

References:

Maina, F. Z., Rosen, D., Abbaszadeh, P., Yang, C., Kumar, S. V., Rodell, M., & Maxwell, R. (2025). Integrating the interconnections between groundwater and land surface processes through the coupled NASA Land Information System and ParFlow environment. Journal of Advances in Modeling Earth Systems (JAMES), 17(2). https://doi.org/10.1029/2024MS004415

Line 134: Change to: ". Rather, it..."

Done.

Line 136: Change to: "Allows a realistic representation of..., to be obtained."

We revised the sentence as you suggested.

*"This inclusive methodology allows a realistic representation of groundwater dynamics to be obtained, shaped by the underlying geology and topography."*

Line 138: As a reader, I am wondering why another approach to model surface processes needs to be proposed?

We used the ParFlow subsurface hydrology model, which solves the partial differential equations (PDEs) governing both surface water and subsurface flow.

Line 143: Change to: "Which ensures?"

Done.

Line 147: It will only "replicate" it if the geometry, material properties, and forcing functions are reasonable. I would prefer a different term like "calculates."

We have revised the sentence as you suggested. Thank you.

Line 149: This is not very clear. I think you are referring to the complexity of the representation of subsurface processes, i.e. the equations. But "subsurface configurations(s)" could also be taken to mean the geology encountered in any given application. There are many factors affecting spin-up time (by the way, why a steady-state solution has to be sought initially in the case of subsequent transient runs has not yet been explained). For instance, it depends on the size of the domain (in 3D) and especially how reasonable the initial conditions applied are. Meanwhile, the domain size, grid resolution, geological complexity, and forcing characteristics also affect the size of the computational problem. I would suggest this part is slightly revised accordingly.

We revised the sentence as you suggested.

"It is important to note that groundwater may take a longer time (compared to soil moisture) to reach steady-state due to the complexity of the representation of subsurface processes. This makes it a computationally intensive problem to solve. There are many factors that influence spin-up time, including the size of the domain (in 3D), the resolution of the grid, the geological complexity, and the characteristics of the forcing data. A steady-state solution is typically sought initially in order to ensure that the model starts from a physically realistic and stable state before transitioning to transient runs. Additionally, the plausibility of the initial conditions applied plays a significant role in determining how quickly the model reaches steady-state. All of these factors contribute to the overall computational complexity of the problem."

Line 154: But these key inputs (initial conditions, boundary conditions, and parameters) also apply for any numerical model, including ParFlow. I find it slightly strange how they are presented here as being somehow unique to LIS.

We revised the sentence as you suggested.

"Land surface modeling within LIS relies on three key inputs: (1) initial conditions, describing the land surface's starting state; (2) boundary conditions, encompassing the atmospheric fluxes or 'forcings' (upper boundary condition) and soil fluxes or states (lower boundary condition); and (3) parameters, which represent the soil, vegetation, topography, and other land surface characteristics. These inputs are not unique to LIS and also apply to other numerical models, such as ParFlow, which similarly requires initial conditions, boundary conditions, and parameters to simulate subsurface and surface processes."

Line 158: Ok, so this addresses the previous comment about the relationship. You could mention earlier that LIS is a kind of model toolbox, with various model options inside?

We have already addressed this comment above.

Line 168: The possibility of performing?

We have revised this sentence as you suggested.

"The DA embedded within LIS provides the possibility of performing probabilistic simulations..."

Line 175: Exfiltration to streams and rivers, right?

Line 176: Could it be useful to also mention how spatial patterns in subsurface properties are represented under this simple approach?

While Noah-MP captures some spatial variability through parameterized subsurface properties, it does not explicitly simulate lateral groundwater flow or fully resolve spatial heterogeneity at fine scales, which can limit accuracy in complex hydrogeological settings. To address these two comments, we revised the text, please see it below:

"This method tracks variations in groundwater storage based on inflow, known as recharge, and outflows, which include capillary rise and exfiltration to streams and rivers (baseflow). It is important to note that this approach does not explicitly consider complex hydraulic properties, such as hydraulic conductivity, which is typically used in soil moisture modeling and groundwater recharge prediction. Additionally, while Noah-MP captures some spatial variability through parameterized subsurface properties, it does not explicitly simulate lateral groundwater flow or fully resolve spatial heterogeneity at fine scales, which can limit accuracy in complex hydrogeological settings. As a result, spatial patterns in subsurface properties are not explicitly represented in this simplified approach, which may influence the accuracy of groundwater storage estimates."

Line 180: I know it was not done and would therefore be another study, but would an interesting comparison not have been to compare ParFlow-Noah-MP with ParFlow-CLM?

Yes, this will certainly be one of our future studies. Since this paper focuses on the details and capabilities of the newly developed coupled subsurface-surface hydrology model (ParFlow-LIS/Noah-MP), we have not included or compared other models. However, exploring this comparison will be an important direction for our future research.

Line 181: Did you couple *all* LIS models, or only Noah-MP? If you also coupled CLM, then how does this differ from the previous coupling? (It seems that this would come down to the ability to do DA, which LIS provides?)

No. We have only coupled LIS/Noah-MP with ParFlow. ParFlow and CLM have already been coupled and used in several other studies.

Line 183: # Or runs off? But runoff is also a land surface process. Which model is this simulated using?

ParFlow takes care of the runoff estimation. LIS only calculates the net water flux entering the soil.

Line 184, 185: Specify ParFlow subsurface model.

Revised.

"... used as input to feed the ParFlow subsurface model."

Line 186, 187: Could be rephrased for greater clarity, e.g. "...top four soil layers: the coupled soil zone, in which the two systems communicate."

Revised.

"It should be noted that the land surface model (LIS/Noah-MP) and groundwater model (ParFlow) share the top four soil layers: the coupled soil zone, in which the two models exchange fluxes."

Also, "in which the two models exchange fluxes" might be better than "communicate"?

Addressed above. Thank you so much for reading our paper so carefully and providing super useful and constructive comments. I really appreciate it.

Finally, I would suggest referring immediately to Figure 1 here, which is useful.

Done.

**"(See Figure 1)"**

Line 192, 193: This is confusing. You say that LIS/Noah calculates the soil moisture, but then next you say this is derived from ParFlow. So is the coupling actually that LIS/Noah-MP calculates recharge to ParFlow, and ParFlow passes back soil moisture content?

Thank you for the precise comment. We have revised this sentence.

*"By using saturation data generated by ParFlow as one of its outputs and incorporating the soil layer porosity values, the soil moisture content (\theta) is estimated."*

Line 204: I guess soil or more generally geological facies-specific storage and porosity? (Below a few meters, there is no soil.)

We have revised this sentence as you suggested.

"ParFlow provides estimates of pressure head and soil saturation, which, along with soilor more generally geological facies-specific storage and porosity, are used to calculate subsurface storage."

Line 208: So lateral subsurface flow is accounted for in this approach, but lateral surface flow is not? Perhaps specify this if necessary, because so far, there has been little mention of how direct surface runoff is handled.

Thank you for the comment. This is also one of the reviewer #1 suggestion and we have added the following append to the revised manuscript to elaborate more on the ParFlow and how it handles direct surface runoff.

"To address this comment, we have added an appendix to the revised manuscript.

**"Appendix**

The ParFlow model operates in three distinct modes: (1) variably saturated; (2) steady-state saturated; and (3) integrated watershed flows. This adaptability enhances its utility across a range of hydrological scenarios. Here we summarize each mode following the work of Kollet and Maxwell (2006).

**Variably Saturated Flow**

ParFlow can operate in variably saturated mode through the well-known mixed form of the Richards' equation:

 $S_{s}S_{w}(p)\frac{\partial p}{\partial t} + \phi \frac{\partial (S_{w}(p))}{\partial t} = \nabla q + q_{s}$ (1)  $q = -k_{s}k_{r}(p)\nabla (p - z)$ (2)

where  $S_s$  is the specific storage coefficient [L-1],  $S_w$  is the relative saturation [-] as a function of pressure head p, t is time,  $\phi$  is the porosity of the medium [-], q is the specific volumetric (Darcy) flux [LT-1],  $k_s$  is the saturated hydraulic conductivity tensor [LT-1],  $k_r$  is the relative permeability [-], which is a function of the pressure head p,  $q_s$  is the general source or sink term [T-1] (includes wells and surface fluxes, e.g., evaporation and transpiration). z represents depth below the surface [L]. ParFlow has been utilized for numerical simulations, including the modeling of river-aquifer exchange involving both free-surface flow and subsurface flow. It has also demonstrated efficacy in addressing highly heterogeneous problems under variably saturated flow conditions. For the situations where the saturated conditions are predominant, the steady-state saturated mode in ParFlow becomes a valuable tool.

**Steady-State Saturated Flow**

The fully saturated groundwater flow equation is expressed as follows:

 $\nabla q - q = 0 \qquad (3)$  $q = -k_{s} \nabla P \qquad (4)$

where *P* represents the 3-D hydraulic head-potential [L]. ParFlow does include a direct solution option for the steady-state saturated flow that is distinct from the transient solver. When studying more sophisticated or complex processes, such as when simulating a fully coupled system is of interest (i.e., surface and subsurface flow), an overland flow boundary condition is employed.

**Overland Flow**

Surface water systems are interlinked with the subsurface system; this interaction plays a critical role for rivers. However, explicitly representing the connections between the two systems in numerical simulations is a difficult task. In ParFlow, overland flow is implemented as a two-dimensional kinematic wave equation approximation of the shallow water equations. The continuity equation for two-dimensional shallow overland flow is expressed as follows:

$$\frac{\partial \psi_s}{\partial t} = \nabla \left( \upsilon \psi_s \right) + q_s \qquad (5)$$

where  $\upsilon$  is the depth-averaged velocity vector [LT–1] and  $\psi_s$  is the surface ponding depth [L]. Ignoring the dynamic and diffusion terms results in the momentum equation, which is known as the kinematic wave approximation:

$$S_{fi} = S_{oi} \quad (6)$$

The  $S_{f,i}^{p,i}$  and  $S_{o,i}^{p,i}$  represent the friction [-] and bed slopes (gravity forcing term) [-], respectively. *i* indicates the *x* and *y* directions in the following equations. Therefore, Manning's equation can be used to build a flow depth-discharge relationship as follows:

$$\upsilon_{x} = \frac{\sqrt{S_{fx}}}{n} \Psi_{s}^{2/3} \qquad (7)$$
$$\upsilon_{y} = \frac{\sqrt{S_{fy}}}{n} \Psi_{s}^{2/3} \qquad (8)$$

where *n* is the Manning roughness coefficient [TL-1/3]. The shallow overland flow formulation (Eq. 9) assumes the vertical averaging of flow depth and disregards any vertical change in momentum within the surface water column. To incorporate vertical flow (from the surface to the subsurface or vice versa), a formulation that couples the system of equations through a boundary condition at the land surface becomes essential. We can modify Equation (5) to include an exchange rate with the subsurface,  $q_e$ :

 $\frac{\partial \psi_s}{\partial t} = \nabla \left( \upsilon \psi_s \right) + q_s + q_e \qquad (9)$

In ParFlow, the overland flow equations are directly coupled to the Richards' equation at the top boundary cell under saturated conditions. Conditions of pressure continuity (i.e., equal pressures at the ground surface for the subsurface and surface domains) and flux at the top cell of the boundary between the subsurface and surface systems are assigned. Setting pressure head in Equation (1) equal to the vertically averaged surface pressure,  $\psi_c$ :

 $p = \Psi_{s} = \Psi \qquad (10)$

and the flux,  $q_{e'}$  equal to the specified boundary conditions (for example, Neumann-type boundary conditions):

 $q_{BC} = -k_s k_r \nabla(\psi - z)$  (11) and one solves for the flux term in Equation (10), the result becomes:

 $q_e = \frac{\partial \|\psi, 0\|}{\partial t} - \nabla v \|\psi, 0\| - q_s \qquad (12)$

where the  $\|\psi, 0\|$  operator is defined as the greater of the quantities,  $\psi$ , 0. Putting the equations (10) and (11) together results in the following relationship:

$$-k_{s}k_{r}\nabla(\psi - z) = \frac{\partial ||\psi,0||}{\partial t} - \nabla \upsilon ||\psi,0|| - q_{s}$$
(13)

As we see here the surface water equations are represented as a boundary condition to the Richards' equation. For more information about the coupled surface and subsurface flow systems in ParFlow, we refer the interested readers to Kollet and Maxwell (2006)."

**References:**

Kollet, S. J. and Maxwell, R. M.: Integrated surface-groundwater flow modeling: A free-surface overland flow boundary condition in a parallel groundwater flow model, Adv Water Resour, 29, https://doi.org/10.1016/j.advwatres.2005.08.006, 2006."

Line 209: Suggest "bi-directional exchange."

**Revised.**

"...facilitating the bi-directional exchange between the land surface and subsurface..."

Line 223: Does this mean that between winters, the maximum SCA fluctuates between these values, or does it mean that within winters the SCA never drops lower than 50,000 and always reaches 280,000 (entire basin)? I guess the former, but as currently written, it is quite ambiguous.

Yes, the former is correct, which means between different winters, the maximum snow-covered area (SCA) varies between 50,000 km2 and 280,000 km2—not that every winter follows the exact same range. We have revised this sentence for further clarity.

"The maximum snow-covered area within the UCRB varies between 50,000 km2 and 280,000 km2 across different winters during the October–April season."

Line 232: Change to: "Including its climatology and geology" (delete etc.).

We have revised this sentence as you suggested.

"For more information about the UCRB, including its climatology and geology, we refer interested readers to Miller et al. (2016)."

Line 239: So are they not used for any calibration/DA? (In general, I also prefer "evaluation" to "validation.")

No. We have revised the sentence as you suggested.

*"In this section we describe all those in-situ observations and satellite products that are used for evaluation of model simulations."*

Line 264: So, is this the second satellite product used? Could perhaps be "signposted" a bit more clearly to readers.

Yes. We have revised this sentence for further clarity.

"The second satellite product that we used in this study for evaluation of coupled PF-LIS model simulation is Anomalies of Terrestrial Water Storage (TWS), derived from the Gravity Recovery and Climate Experiment (GRACE; Tapley et al., 2004) and its successor, GRACE Follow-On (GRACE-FO; Landerer et al., 2020).

Line 280: It would be useful to show the river network too on the left, so that readers can understand the orders of the streams gauged, etc.

To address this comment, we have added the following figure to the revised supplementary file. Thank you for the suggestion.

*Figure S7. UCRB along with USGS streamflow stations and river networks.*

Line 292: Resampled. Using which algorithm?

Using the nearest neighbor method. We have revised that sentence and included this information.

"Land cover information was extracted from the National Land Cover Database (NLCD) at a 30-meter resolution and subsequently resampled using the nearest neighbor method to match the model's 1-kilometer resolution."

Line 301, 302: Please explain more. Which parameters are detailed in which paper? Without this info, there is a risk that the study would not be reproducible.

We have revised the text and included more information.

"The development of the 3D subsurface, which includes soil datasets (e.g., permeability and porosity), unconsolidated, a semi-confining layer, bedrock aquifers, and the 3D model grid, is detailed in Tijerina-Kreuzer et al (2024). The subsurface parameters (e.g. saturated hydraulic conductivity and van Genuchten parameters for the soil and subsurface) are detailed in Yang et al (2023)."

Line 306: What resolution and timestep?

We have revised the text and included this information.

"For the atmospheric forcing data, we use the phase-2 of the North American Land Data Assimilation System (NLDAS-2) product (https://ldas.gsfc.nasa.gov/nldas/v2/forcing). This dataset, available at a spatial resolution of 12.5 km and a temporal resolution of hourly, includes eight variables: precipitation, air temperature, shortwave and longwave radiation, wind speed in two directions (east-west and south-north), atmospheric pressure, and specific humidity."

Line 312: I would rather say "reasonable initial conditions." Line 313: You say above that you had to spin up ParFlow, but here you say you simply took existing output. Please revise for consistency, e.g. "Reasonable initial conditions had to be obtained for both models. To do this, ..."

To address the above two comments, we have revised the text as you suggested.

"Reasonable initial conditions had to be obtained for both models. To do this, The initial condition (i.e., pressure head) for the ParFlow model was directly obtained from Yang et al (2023). who spunup the ParFlow model over the entire CONUS."

Line 327: Consider adding "..., relative to LIS/Noah-MP alone."

We have revised the text as you suggested.

"...how the coupled system can enhance the modeling of land surface processes and provide a more accurate representation of groundwater storage, relative to LIS/Noah-MP alone.

Line 334: Extending to.

Corrected.

Line 349: I think you mean spatial resolution or specificity.

We have revised the text as you suggested.

*"PF-LIS/Noah-MP provides soil moisture data with higher spatial specificity, which can be..."*

Line 352: Essentially, it seems that the version with ParFlow is able to "pick out" areas of high soil moisture along river corridors. Can you suggest why this could be? E.g. is it all re-exfiltrating groundwater, or can it be explained by any differences between the representation of topography or river channels in the models?

By integrating surface runoff and subsurface flow processes, ParFlow can simulate the lateral movement of water across the landscape. This dynamic simulation accounts for the redistribution of water due to both surface topography and subsurface properties, leading to a more accurate depiction of soil moisture patterns, particularly in regions influenced by river networks. In addition, the model utilizes high-resolution topographic data to define the land surface and river channels accurately. This precise representation allows ParFlow to identify topographic depressions and convergent zones where water is likely to accumulate, leading to higher soil moisture content. Such detailed modeling ensures that areas prone to saturation, especially along river corridors, are effectively captured.

To further address this comment, we included the above information in the revised text to highlight why the coupled system is able to "pick out" areas of high soil moisture along river corridors.

"The results (shown in Figure 4) also reveal that the coupled system is able to identify the areas of high soil moisture along the river corridors. This is attributed to the ParFlow model. By integrating surface runoff and subsurface flow processes, ParFlow can simulate the lateral movement of water across the landscape. This dynamic simulation accounts for the redistribution of water due to both surface topography and subsurface properties, leading to a more accurate depiction of soil moisture patterns, particularly in regions influenced by river networks. In addition, the model utilizes high-resolution topographic data to define the land surface and river channels accurately. This precise representation allows ParFlow to identify topographic depressions and convergent zones where water is likely to accumulate, leading to higher soil moisture content. Such detailed modeling ensures that areas prone to saturation, especially along river corridors, are effectively captured."

Line 380, 381: Yes, it is interesting that when aggregated, there is little difference in the fit scores. In one way, this illustrates the limitation, given their limited resolution, of satellite products to evaluate high-resolution simulations.

Yes, this is true. This is an inevitable issue when using satellite products for model evaluation.

Line 399: Elevations.

Corrected.

Line 404: So can LIS/Noah-MP not do this? If so, it would be worth explaining in an earlier section.

Noah-MP can model soil moisture dynamics, but there are limitations in handling complex topography and boundary conditions compared to ParFlow. Noah-MP primarily uses a structured grid system, and while it allows for parameterization of soil moisture processes, it does not inherently solve three-dimensional flow dynamics as ParFlow does. This means that in regions with steep gradients or highly heterogeneous terrain, ParFlow is better suited because it explicitly resolves lateral flow and subsurface heterogeneity. ParFlow's numerical formulation allows for more flexible and realistic boundary condition representations, particularly in irregular terrains. Noah-MP, on the other hand, relies on predefined lookup tables and soil parameterizations that may not fully capture spatial heterogeneity. Noah-MP can incorporate variations in soil moisture, but its hydraulic conductivity representation is more generalized, whereas ParFlow enables explicit simulation of hydraulic gradients and subsurface interactions.

To further address this comment, we have added the above text to the revised manuscript.

"Noah-MP can model soil moisture dynamics, but there are limitations in handling complex topography and boundary conditions compared to ParFlow. Noah-MP primarily uses a structured grid system, and while it allows for parameterization of soil moisture processes, it does not inherently solve three-dimensional flow dynamics as ParFlow does. This means that in regions with steep gradients or highly heterogeneous terrain, ParFlow is better suited because it explicitly resolves lateral flow and subsurface heterogeneity. ParFlow's numerical formulation allows for more flexible and realistic boundary condition representations, particularly in irregular terrains. Noah-MP, on the other hand, relies on predefined lookup tables and soil parameterizations that may not fully capture spatial heterogeneity. Noah-MP can incorporate variations in soil moisture, but its hydraulic conductivity representation is more generalized, whereas ParFlow enables explicit simulation of hydraulic gradients and subsurface interactions." Line 405, 406: Please compare with what the LIS/Noah-MP does.

Noah-MP does account for both capillary rise and gravitational effects, but its treatment differs from ParFlow in terms of complexity and implementation. Noah-MP incorporates capillary rise through its soil moisture parameterization, which includes the influence of matric potential on soil water movement. However, its representation is more simplified and dependent on predefined soil layers, limiting its ability to dynamically capture variations in soil moisture redistribution, particularly in highly heterogeneous terrains. Noah-MP includes gravitational drainage as part of its hydrology scheme, but it assumes a one-dimensional vertical flow, meaning lateral subsurface flow and complex topographic-driven gravitational effects are not fully resolved. In contrast, ParFlow explicitly solves the three-dimensional Richards' equation, which allows it to capture both vertical and lateral water movement more accurately in complex terrains.

To further address this comment, we have added the above text to the revised manuscript.

"Noah-MP accounts for both capillary rise and gravitational effects, however, its treatment differs from ParFlow in terms of complexity and implementation. Noah-MP incorporates capillary rise through its soil moisture parameterization, which includes the influence of matric potential on soil water movement. However, its representation is more simplified and dependent on predefined soil layers, limiting its ability to dynamically capture variations in soil moisture redistribution, particularly in highly heterogeneous terrains. Noah-MP includes gravitational drainage as part of its hydrology scheme, but it assumes a one-dimensional vertical flow, meaning lateral subsurface flow and complex topographic-driven gravitational effects are not fully resolved. In contrast, ParFlow explicitly solves the three-dimensional Richards' equation, which allows it to capture both vertical and lateral water movement more accurately in complex terrains."

Line 413: Since you said that the coupled model simulations are limited to the boundaries of the UCRB, I am struggling to interpret this figure. Would it not make sense to remove points outside the catchment? It is also quite difficult to see and compare any differences, apart from via the average statistics.

To address this comment, we have decided to reproduce all these four figures. Thank you for the suggestion.

---

## Author Comment (AC3)

**Reviewer #3**

The authors present a case study on a coupled modeling system that integrates ParFlow with LIS/Noah-MP to simulate land surface and subsurface hydrologic processes in the Upper Colorado River Basin. They compare the LIS/Noah-MP and PF-LIS/Noah-MP model simulations with in-situ and satellite observations of soil moisture, streamflow, water table depth, and terrestrial water storage. The paper is well-structured and well-illustrated, offering a significant contribution to the field. However, I recommend major revisions to improve clarity and improve the overall quality of the presentation.

Thank you for the useful comments and suggestions. These insights have significantly enhanced both the quality and clarity of our work, helping us refine key aspects of the study and improve its overall readability. We appreciate the time and effort the reviewer dedicated to providing detailed feedback, which has been instrumental in strengthening the presentation and impact of our paper.

**Major comments:**

The abstract and conclusion do not sufficiently address the motivation and novelty of the research. While the study presents a coupled modeling system, it is unclear what key advancements or unique contributions it offers compared to existing methods. For example, why was ParFlow chosen to represent hydrological processes? Why was LIS/Noah-MP used as the land surface model? What advantages does the LIS system offer over using the standalone Noah-MP model? Additionally, while data assimilation within LIS system is mentioned, no corresponding results are presented in this manuscript. If possible, could you at least discuss the potential future advantages of incorporating LIS system? I recommend explicitly stating the research gap this study aims to fill and clearly articulating the novel aspects of the approach in both the abstract and conclusion.

To address this comment, we have revised the abstract and conclusion sections of the manuscript to ensure that all the reviewer's suggestions and comments are thoroughly incorporated.

**"Abstract**

Understanding, observing, and simulating Earth's water cycle is imperative for effective water resource management in the face of a changing climate. While NASA's Land Information System (LIS)/Noah-MP is widely used for land surface modeling, its ability to

represent groundwater processes is limited. In contrast, the ParFlow hydrologic model explicitly simulates subsurface water movement. This study introduces a newly coupled modeling framework, ParFlow-LIS/Noah-MP (PF-LIS/Noah-MP), which integrates the strengths of both models to provide a physically based representation of surface and subsurface processes and their interactions. Unlike standalone LIS/Noah-MP, the coupled system enables three-dimensional groundwater flow simulations by solving the Richards equation, improving the realism of subsurface hydrologic processes.

We evaluate PF-LIS/Noah-MP over the Upper Colorado River Basin (UCRB) by comparing its simulations against in-situ and satellite observations, including soil moisture, streamflow, and groundwater storage. In general, the results show that PF-LIS/Noah-MP produces soil moisture simulations comparable to those of LIS/Noah-MP across the entire UCRB, with nearly identical root mean squared error and correlation coefficients. However, further analysis—when these metrics are averaged over areas with complex topography—revealed that in regions with high elevation gradients, PF-LIS/Noah-MP outperforms standalone LIS/Noah-MP in soil moisture simulation. The coupled model's ability to simulate groundwater storage and lateral subsurface flow introduces new hydrologic prediction capabilities that were not possible within the standalone LIS/Noah-MP model."

**"Conclusion**

In this study, we introduced a coupled surface-subsurface hydrology model, *PF-LIS/Noah-MP*, and studied its performance in estimating different hydrologic variables. This study was conducted in the UCRB, a region heavily dependent on groundwater to supply water for millions of people in the western United States. With an anticipated increase in drought occurrences due to climate warming, the region faces a heightened risk of groundwater depletion in the future. Understanding the dynamics of land surface and subsurface water in the UCRB is crucial for effective water resource management and policymaking.

In this study, we employed the recently developed integrated surface-subsurface hydrology model, PF-LIS/Noah-MP, to assess key components such as soil moisture, streamflow, water table depth, and total water storage anomaly across the UCRB. These estimations were then compared with a comprehensive set of in-situ and satellite observations, encompassing soil moisture data from various networks, USGS streamflow and well observations, as well as satellite data from SMAP for soil moisture and GRACE for groundwater.

The findings demonstrate that the integration of ParFlow with LIS/Noah-MP expands the physics represented by the LIS/Noah-MP model. These increased process representations have two main advantages: better performance of land surface fluxes, especially in regions with complex topography, and accurate estimations of subsurface hydrologic processes, including water table depth. In particular, our results highlight that the coupled PF-LIS/Noah-MP model improves soil moisture representation in steep terrain, where standalone LIS/Noah-MP struggles due to its simplified groundwater formulation. This enhanced performance is crucial for capturing water availability in headwater regions, which serve as critical water sources for downstream users. Moreover, the ability to simulate lateral subsurface flow offers an improved understanding of groundwater redistribution, an important mechanism influencing baseflow and long-term water availability.

PF-LIS/Noah-MP presents a viable approach to studying land surface and subsurface hydrologic processes and their interactions across different scales. This research contributes valuable insights for informed decision-making in the management of water resources in the UCRB, particularly in the face of future climate challenges. The ability of PF-LIS/Noah-MP to explicitly resolve groundwater processes also makes it a promising tool for evaluating the impacts of future climate scenarios on water availability, particularly in arid and semi-arid regions where groundwater plays a crucial role in sustaining ecosystems and human activities. Future work should explore the model's sensitivity to different parameterizations and meteorological forcing datasets, which could further refine its applicability for large-scale hydrologic assessments.

Although the current study does not explicitly incorporate groundwater pumping or *irrigation, these processes are essential for understanding regional water dynamics. The* observed discrepancies between PF-LIS/Noah-MP groundwater simulations and *GRACE-derived groundwater storage highlight the need to account for human impacts* on groundwater availability. Future work can leverage data assimilation techniques to integrate observed groundwater data and improve model accuracy. The more detailed representation of subsurface processes within the PF-LIS/Noah-MP system allows for improved utilization of remote sensing information through data assimilation. For example, to date, the assimilation of GRACE terrestrial water storage observations has only been demonstrated within models that have a shallow groundwater representation and without the representation of lateral subsurface moisture transport processes (e.g., Kumar et al., 2016). By incorporating a fully integrated subsurface representation, *PF-LIS/Noah-MP offers an opportunity to advance hydrologic data assimilation systems* by directly leveraging GRACE-based water storage estimates. The ongoing development will extend LIS' data assimilation capabilities to PF-LIS, to enable better exploitation of the information from remote sensing."

The description of the coupled modeling system lacks sufficient detail for a clear understanding. The explanation heavily relies on citations, including unpublished material (e.g., Fadji et al., 2024), which may limit accessibility to critical information. I recommend reconsidering the citation of unpublished sources and providing a more comprehensive description of the ParFlow-LIS/Noah-MP coupled system to better highlight its strengths.

This paper by Maina et al. (2025) was recently published in the Journal of Advances in Modeling Earth Systems. Here is the link to the paper: https://doi.org/10.1029/2024MS004415. We have updated the references in the revised manuscript.

**References**

Maina, F. Z., Rosen, D., Abbaszadeh, P., Yang, C., Kumar, S. V., Rodell, M., & Maxwell, R. (2025). Integrating the interconnections between groundwater and land surface processes through the coupled NASA Land Information System and ParFlow environment. Journal of Advances in Modeling Earth Systems (JAMES), 17(2). https://doi.org/10.1029/2024MS004415

We have incorporated the following text into Section 4 to further emphasize the details of the coupled system.

*"The LIS/Noah-MP model is designed to simulate the energy and water fluxes at the land"* surface, along with key state variables like ET and its components, snow-related variables (such as SWE and snow cover), and infiltration. It computes the surface energy balance by representing vegetation with a detailed canopy model, incorporating its dimensions, orientation, density, and radiometric properties. A two-stream radiation transfer scheme is employed to account for the complex interactions of solar radiation within the canopy. For snow processes, the model features a multi-layer snowpack, capable of storing liquid water and simulating melt and refreeze processes. It also includes a snow interception component, which models the loading and unloading of snow, sublimation, and other snow-related processes. The ET and infiltration values (which combine snowmelt and rainfall) produced by LIS are passed on to ParFlow. ParFlow then calculates the surface, soil, and subsurface hydrodynamics, generating important hydrological outputs such as water table depth, groundwater storage (derived from pressure-head and saturation), soil moisture, and streamflow (Maina et al., 2025). In particular, transpiration is computed by LIS/Noah-MP using the soil moisture computed by ParFlow. Within LIS/Noah-MP, transpiration is computed using a Penman-Monteith based approach, where stomatal resistance (influenced by solar

radiation, vapor pressure deficit, temperature, and soil moisture) controls canopy conductance. Actual transpiration is obtained by scaling potential transpiration with a soil moisture stress function, considering vegetation type, root distribution, and dynamic LAI."

To further address this comment, we have also added an appendix to the revised manuscript. This provides more information about how the subsurface processes are simulated within the coupled system and its strengths.

**"Appendix**

The ParFlow model operates in three distinct modes: (1) variably saturated; (2) steady-state saturated; and (3) integrated watershed flows. This adaptability enhances its utility across a range of hydrological scenarios. Here we summarize each mode following the work of Kollet and Maxwell (2006).

**Variably Saturated Flow**

ParFlow can operate in variably saturated mode through the well-known mixed form of the Richards' equation:

$$S_{s}S_{w}(p)\frac{\partial p}{\partial t} + \phi \frac{\partial (S_{w}(p))}{\partial t} = \nabla q + q_{s}$$
(1)
$$q = -k_{s}k_{r}(p)\nabla (p - z)$$
(2)

where  $S_s$  is the specific storage coefficient [L-1],  $S_w$  is the relative saturation [–] as a function of pressure head p, t is time,  $\phi$  is the porosity of the medium [–], q is the specific volumetric (Darcy) flux [LT–1],  $k_s$  is the saturated hydraulic conductivity tensor [LT–1],  $k_r$  is the relative permeability [–], which is a function of the pressure head p,  $q_s$  is the general source or sink term [T–1] (includes wells and surface fluxes, e.g., evaporation and transpiration). z represents depth below the surface [L]. ParFlow has been utilized for numerical simulations, including the modeling of river-aquifer exchange involving both free-surface flow and subsurface flow. It has also demonstrated efficacy in addressing highly heterogeneous problems under variably saturated flow conditions. For the situations where the saturated conditions are predominant, the steady-state saturated mode in ParFlow becomes a valuable tool.

**Steady-State Saturated Flow**

The fully saturated groundwater flow equation is expressed as follows:

 $\nabla q - q = 0$  (3)  $q = -k_s \nabla P$  (4)

where *P* represents the 3-D hydraulic head-potential [L]. ParFlow does include a direct solution option for the steady-state saturated flow that is distinct from the transient solver. When studying more sophisticated or complex processes, such as when simulating a fully coupled system is of interest (i.e., surface and subsurface flow), an overland flow boundary condition is employed.

**Overland Flow**

Surface water systems are interlinked with the subsurface system; this interaction plays a critical role for rivers. However, explicitly representing the connections between the two systems in numerical simulations is a difficult task. In ParFlow, overland flow is implemented as a two-dimensional kinematic wave equation approximation of the shallow water equations. The continuity equation for two-dimensional shallow overland flow is expressed as follows:

$$\frac{\partial \Psi_s}{\partial t} = \nabla \left( \upsilon \Psi_s \right) + q_s \qquad (5)$$

where  $\upsilon$  is the depth-averaged velocity vector [LT–1] and  $\psi_s$  is the surface ponding depth [L]. Ignoring the dynamic and diffusion terms results in the momentum equation, which is known as the kinematic wave approximation:

$$S_{f,i} = S_{o,i} \quad (6)$$

The  $S_{f,i}$  and  $S_{o,i}$  represent the friction [-] and bed slopes (gravity forcing term) [-], respectively. *i* indicates the *x* and *y* directions in the following equations. Therefore, Manning's equation can be used to build a flow depth-discharge relationship as follows:

$$\upsilon_{x} = \frac{\sqrt{S_{fx}}}{n} \psi_{s}^{2/3} \qquad (7)$$
$$\upsilon_{y} = \frac{\sqrt{S_{fy}}}{n} \psi_{s}^{2/3} \qquad (8)$$

where *n* is the Manning roughness coefficient [TL-1/3]. The shallow overland flow formulation (Eq. 9) assumes the vertical averaging of flow depth and disregards any vertical change in momentum within the surface water column. To incorporate vertical flow (from the surface to the subsurface or vice versa), a formulation that couples the system of equations through a boundary condition at the land surface becomes essential. We can modify Equation (5) to include an exchange rate with the subsurface,  $q_e$ :

$$\frac{\partial \psi_s}{\partial t} = \nabla \left( \upsilon \psi_s \right) + q_s + q_e \qquad (9)$$

In ParFlow, the overland flow equations are directly coupled to the Richards' equation at the top boundary cell under saturated conditions. Conditions of pressure continuity (i.e., equal pressures at the ground surface for the subsurface and surface domains) and flux at the top cell of the boundary between the subsurface and surface systems are assigned. Setting pressure head in Equation (1) equal to the vertically averaged surface pressure,  $\psi_{a}$ :

 $p = \Psi_s = \Psi \qquad (10)$

and the flux,  $q_{e'}$  equal to the specified boundary conditions (for example, Neumann-type boundary conditions):

 $q_{BC} = -k_s k_r \nabla(\psi - z) \qquad (11)$

and one solves for the flux term in Equation (10), the result becomes:

 $q_{e} = \frac{\partial \|\psi, 0\|}{\partial t} - \nabla v \|\psi, 0\| - q_{s} \qquad (12)$

where the  $\|\psi, 0\|$  operator is defined as the greater of the quantities,  $\psi$ , 0. Putting the equations (10) and (11) together results in the following relationship:

$$-k_{s}k_{r}\nabla(\psi-z) = \frac{\partial\|\psi,0\|}{\partial t} - \nabla \upsilon\|\psi,0\| - q_{s}$$
(13)

As we see here the surface water equations are represented as a boundary condition to the Richards' equation. For more information about the coupled surface and subsurface flow systems in ParFlow, we refer the interested readers to Kollet and Maxwell (2006)."

**References:**

Kollet, S. J. and Maxwell, R. M.: Integrated surface-groundwater flow modeling: A free-surface overland flow boundary condition in a parallel groundwater flow model, Adv Water Resour, 29, https://doi.org/10.1016/j.advwatres.2005.08.006, 2006.

The descriptions of the figures are difficult to follow. I recommend explicitly referring to the quantities of model outputs, observations, or evaluation metrics to enhance clarity. This lack of clarity also makes it challenging to follow the conclusion drawn in the paper. For example, the authors state that the coupled modeling system improves simulation performance in regions with complex topography, yet the correlation coefficients and root mean squared errors shown in the figures appear similar. If

possible, please specify which regions show improvements and support this claim with corresponding evaluations metrics.

Over the regions with complex topography (regions with high elevation gradient – shown in Figure 2), the ParFlow-LIS/Noah-MP resulted in relatively better model performance in terms of soil moisture simulation compared to standalone LIS/Noah-MP model. To further clarify this point and address the reviewer's comment, we have added two additional figures to the revised manuscript, revised Figure 1, and added the following text to the revised manuscript.

*Figure 2. Topography of the Upper Colorado River Basin (UCRB) and its location in the US. Regions 1 and 2 represent the areas with complex topography - regions with high elevation